# The non-coding RNA *BC1* regulates experience-dependent structural plasticity and learning

Victor Briz[1,2], Leonardo Restivo [3,4,8], Emanuela Pasciuto[1,2,5], Konrad Juczewski[6,7], Valentina Mercaldo[1,4,8], Adrian C. Lo [1,2,8], Pieter Baatsen[9], Natalia V. Gounko[9], Antonella Borreca[1,2,3,10], Tiziana Girardi[2], Rossella Luca[1,2,3], Julie Nys[1,2,11], Rogier B. Poorthuis[12], Huibert D. Mansvelder [12], Gilberto Fisone[6], Martine Ammassari-Teule[3], Lutgarde Arckens[11], Patrik Krieger[6,13], Rhiannon Meredith [12] & Claudia Bagni [1,2,8,10]

The brain cytoplasmic (*BC1*) RNA is a non-coding RNA (ncRNA) involved in neuronal translational control. Absence of *BC1* is associated with altered glutamatergic transmission and maladaptive behavior. Here, we show that pyramidal neurons in the barrel cortex of *BC1* knock out (KO) mice display larger excitatory postsynaptic currents and increased spontaneous activity in vivo. Furthermore, *BC1* KO mice have enlarged spine heads and postsynaptic densities and increased synaptic levels of glutamate receptors and PSD-95. Of note, *BC1* KO mice show aberrant structural plasticity in response to whisker deprivation, impaired texture novel object recognition and altered social behavior. Thus, our study highlights a role for *BC1* RNA in experience-dependent plasticity and learning in the mammalian adult neocortex, and provides insight into the function of brain ncRNAs regulating synaptic transmission, plasticity and behavior, with potential relevance in the context of intellectual disabilities and psychiatric disorders.

[1] KU Leuven, Department of Neurosciences, Leuven Research Institute for Neuroscience and Disease (LIND), KU Leuven, Leuven 3000, Belgium. [2] VIB Center for Brain & Disease Research, Department of Neurosciences KU Leuven, Leuven 3000, Belgium. [3] Institute of Cell Biology and Neurobiology, CNR, Roma 00015, Italy. [4] Program in Neurosciences and Mental Health, Hospital for Sick Children, Toronto, ON M5G 1X8, Canada. [5] Department of Microbiology and Immunology, KU Leuven, Leuven 3000, Belgium. [6] Department of Neuroscience, Karolinska Institutet, Stockholm 171 77, Sweden. [7] National Institute on Alcohol Abuse and Alcoholism, National Institutes of Health, Rockville, MD 20852, USA. [8] Department of Fundamental Neurosciences, University of Lausanne, Lausanne 1005, Switzerland. [9] VIB Electron Microscopy Platform & Bio Imaging Core, VIB-KU Leuven Center for Brain & Disease Research, Department of Neurosciences KU Leuven, Leuven 3000, Belgium. [10] Department of Biomedicine and Prevention, University of Rome Tor Vergata, Rome 00173, Italy. [11] Department of Biology, Laboratory of Neuroplasticity and Neuroproteomics, KU Leuven, Leuven 3000, Belgium. [12] Department of Integrative Neurophysiology, Center for Neurogenomics and Cognitive Research, VU University, Amsterdam 1081, The Netherlands. [13] Department of Systems Neuroscience, Medical Faculty Bochum, Ruhr-University, Bochum 44801, Germany. Victor Briz, Leonardo Restivo and Emanuela Pasciuto contributed equally to this work. Correspondence and requests for materials should be addressed to C.B. (email: claudia.bagni@unil.ch)

Over the past decade, non-coding RNAs (ncRNAs) have emerged as critical regulatory elements in brain development and function. Different classes of ncRNAs with important roles in synaptic development, maturation and plasticity have been identified, including microRNAs and long ncRNAs[1, 2]. The primate ncRNA *BC200* (Brain Cytoplasmic 200) has been implicated in neurodegenerative and neoplastic processes[2, 3], while the rodent *BC1* RNA was shown to affect susceptibility to audiogenic seizures[4, 5], exploratory behavior, and anxiety[6]. However, the understanding of how these ncRNAs regulate brain function and behavior is still very limited.

*BC1* RNA was initially identified from rat brain[7] and is highly expressed in the cortex[8]. *BC1* RNA localizes in both axons[9] and dendrites[8, 10], where it forms ribonucleoprotein particles (RNPs) with different protein partners, including the Fragile X Mental Retardation Protein (FMRP)[11, 12], poly(A) binding protein 1 (PABP1)[13], the eukaryotic initiation factor 4 A (eIF4A)[14], and eIF4B[15]. Some of these *BC1*-containing RNPs are involved in neuronal translational control[11, 16]. In particular, the FMRP-*BC1*

complex represses translation of a defined subset of FMRP target mRNAs[11, 17] and helps recruit additional factors such as cytoplasmic FMRP interacting protein 1 (CYFIP1), which are responsible for translation inhibition[16]. This function is of particular relevance at synapses[11], where a fine-tuned regulation of local protein synthesis is crucial for spine structure and function[18].

While *BC1* KO mice show no gross abnormalities in regional brain morphology[19], they exhibit abnormal synaptic transmission in the striatum[9, 20, 21], altered neuronal excitability in specific hippocampal regions[4] and mild deficits in place learning[5]. Furthermore, a study performed in an outdoor open field revealed that absence of *BC1* RNA negatively impacts the survival rate[6], suggesting that this ncRNA could contribute to the plastic changes of neuronal connectivity in the neocortex required for the adaptive modulation of behavior. Yet, the involvement of *BC1* RNA in cortical physiology and plasticity is largely unknown.

Here we explore the role of *BC1* RNA in the barrel cortex, a brain area that processes sensory input from the whiskers and is

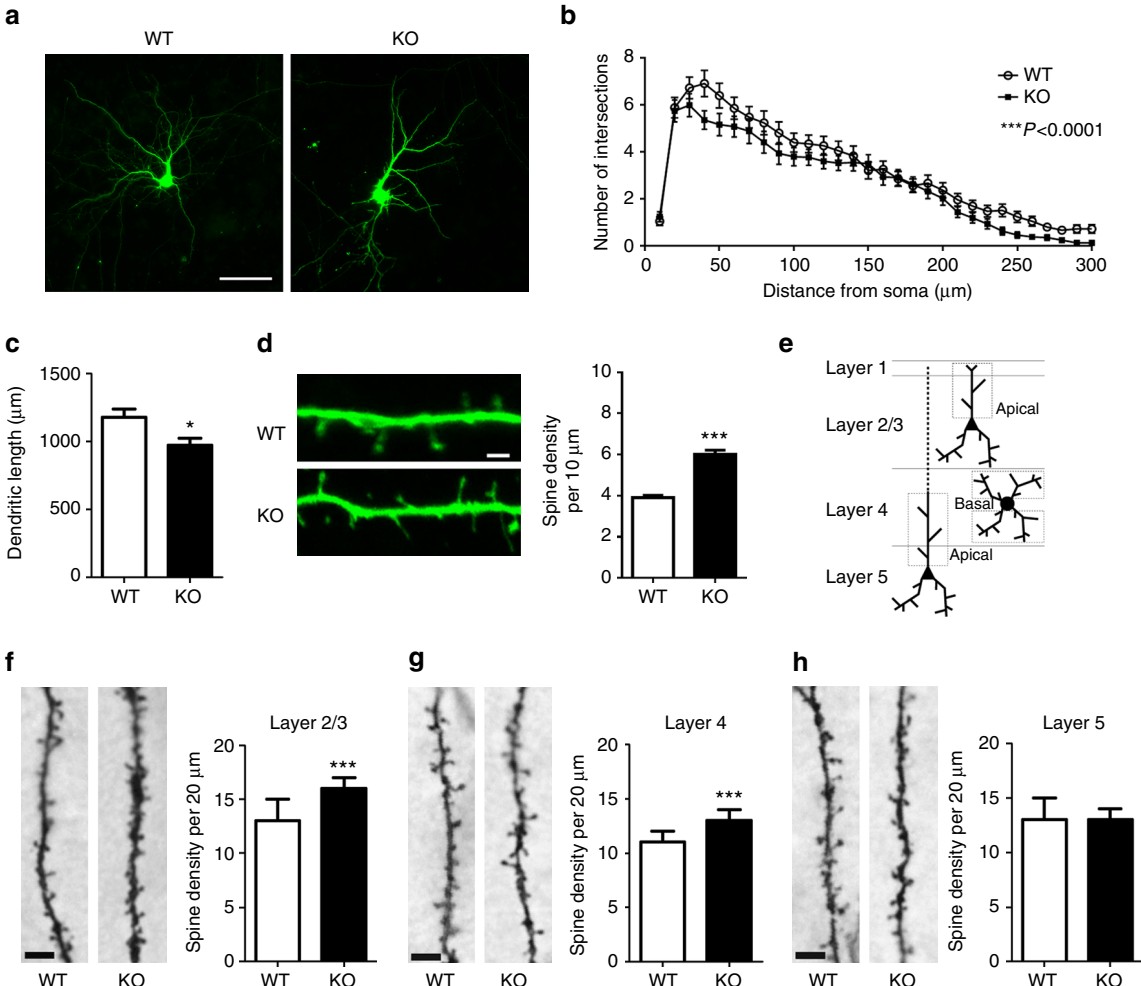

**Fig. 1** *BC1* KO neurons have increased spine density and decreased dendritic complexity. **a** Representative images of WT and *BC1* KO cultured primary neurons transfected with EGFP. *Scale bar*, 100 μm. **b** Sholl analysis and quantification of the number of intersections as a function of distance from the soma (***$P < 0.001$, two-way ANOVA, mean ± s.e.m., $n = 37$ WT and 36 KO neurons). **c** Dendritic length in WT and *BC1* KO neurons (*$P < 0.05$, two-tailed *t*-test, mean ± s.e.m., $n = 37$ WT and 36 KO neurons). **d** *left* Representative images of dendritic segments of neurons transfected with EGFP. *Scale bar*, 2 μm. *right* Quantification of spine density in EGFP-transfected cortical neurons (***$P < 0.001$, two-tailed *t*-test, mean ± s.e.m., $n = 76$ and 68 dendritic segments from WT and KO neurons, respectively). **e** Scheme of the cortical layers and dendrites analyzed in the somatosensory cortex relative to **f–h**. **f–h** *left* In each group, representative images of dendritic segments of neurons from layers 2/3, 4 or 5 used to analyze spine density. *right* Median values ± median absolute deviation (m.a.d.) of spine density distributions (***$P < 0.001$, Mann–Whitney *U*-test; $n = 136/187$, 201/218 and 137/180 dendritic segments from layers 2/3, 4 and 5, respectively, in WT/KO mice). *Scale bar*, 5 μm

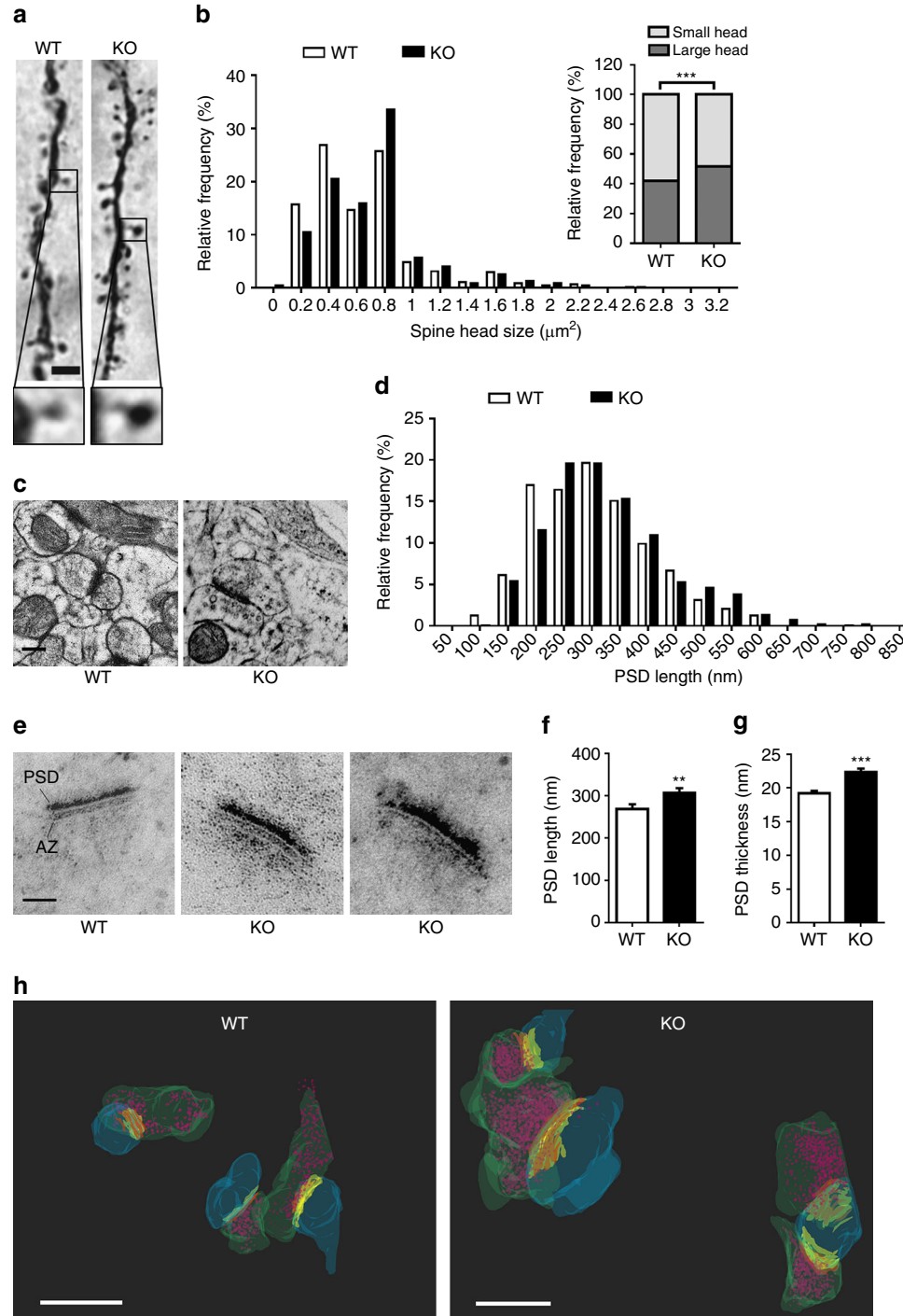

**Fig. 2** *BC1* KO pyramidal neurons have enlarged spine heads and postsynaptic densities. **a** Biocytin-filled dendritic segments from layer 2/3 barrel cortex neurons. *Scale bar*, 2 μm. **b** Frequency distribution of spine head size in dendritic segments ($\chi^2 = 32.99$, df = 1, ***$P < 0.001$, $n = 1303$ and 2747 spines for WT and KO mice, respectively). **c** Representative electron micrographs of spines from the barrel cortex of WT and *BC1* KO mice. *Scale bar*, 250 nm. **d** Frequency distribution of spines as a function of PSD length ($n = 369$ and 615 spines for WT and *BC1* KO mice, respectively). **e** Representative electron micrographs after phosphotungstic acid staining depicting the postsynaptic density (PSD) and active zone (AZ) of spines from the barrel cortex of WT and *BC1* KO mice. *Scale bar*, 100 nm. **f**, **g** Mean ( ± s.e.m.) PSD length and thickness in *BC1* KO compared to WT mice (**$P < 0.01$, ***$P < 0.001$, Mann–Whitney *U*-test, $n = 99$ and 83 spines for WT and KO, respectively). **h** Three-dimensional reconstruction of synapses from the barrel cortex of WT and *BC1* KO mice after SBF-SEM imaging. Dendritic spines (*blue*), PSD (*yellow*), presynaptic terminals (*green*), AZ (*orange*) and synaptic vesicles (*magenta*) are illustrated. *Scale bar*, 500 nm

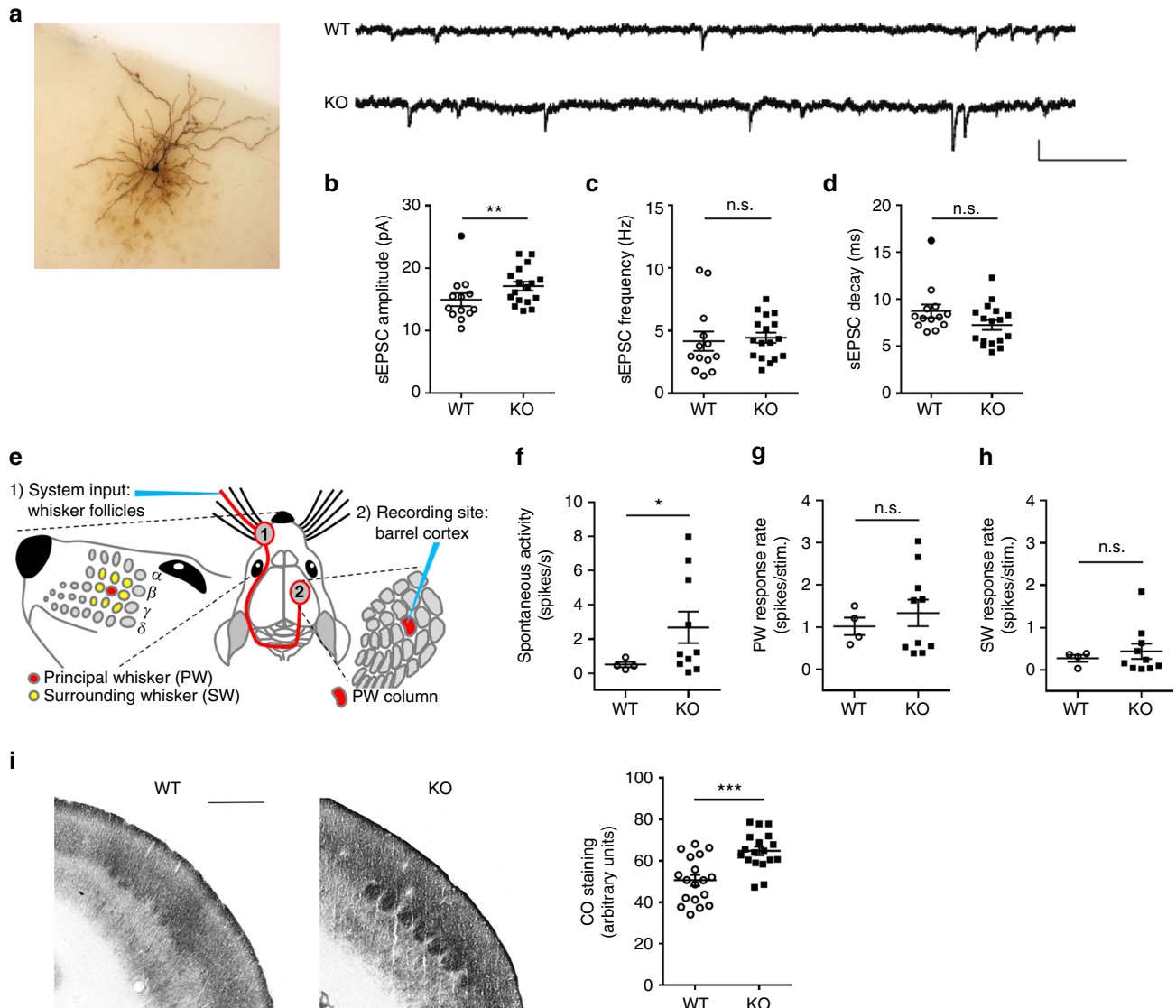

**Fig. 3** *BC1* KO pyramidal neurons display increased spontaneous synaptic activity. **a** Sample traces of spontaneous EPSCs (sEPSC) recorded from layer 2/3 pyramidal neurons of WT and *BC1* KO barrel cortex. *Scale bars*, 20 pA, 100 ms. **b–d** Quantification of sEPSC amplitude, frequency and kinetics (**$P < 0.01$, two-tailed *t*-test, mean ± s.e.m., *n* = 13 and 17 neurons from WT and KO mice, respectively). Outliers from **b** and **d** (*closed circles*) were excluded from the statistical analysis. n.s., not significant. **e** Scheme illustrating the stimulation of principal (PW) and surrounding whiskers (SW). **f** Spontaneous activity in layer 2/3 barrel cortex neurons (*$P < 0.05$, unpaired *t*-test with Welch correction, mean ± s.e.m., *n* = 4 and 10 neurons from WT and KO mice, respectively). **g**, **h** Stimulation of PW and SW and response rates of layer 2/3 pyramidal neurons ($P = 0.4141$ and 0.4201, unpaired *t*-test with Welch correction, mean ± s.e.m., *n* = 4 and 10 neurons from 3 WT and 5 KO mice, respectively). **i** *left* Photomicrographs of cytochrome oxidase (CO)-stained barrels in somatosensory cortex of WT and *BC1* KO mice. Scale bar, 500 μm. *right* Scatter plot of the mean optical density of CO staining in the barrels. *Dots* represent individual barrel values (***$P = 0.001$, Mann–Whitney *U*-test, mean ± s.e.m., *n* = 18 barrels)

well-defined in terms of functional organization and developmental plasticity[22]. In the absence of *BC1* RNA, we observe an increased number of spines and an enlargement of the postsynaptic density (PSD), along with increased spontaneous excitatory postsynaptic responses both ex vivo and in vivo. Furthermore, structural plasticity following sensory input deprivation and behavior requiring sensory processing are impaired in *BC1* KO mice.

## Results

### *BC1* RNA regulates dendritic complexity and spine formation.
Dysregulation of molecules involved in dendritic and spine patterning leads to diverse neuropsychiatric disorders, including Fragile X syndrome (FXS), autism spectrum disorder (ASD) and schizophrenia[23–27]. Given the presence of *BC1* RNA in several

messenger RNPs involved in synaptic development and function, the lack of *BC1* could have a detrimental impact on dendrite elaboration and spine formation. Dendritic complexity was analyzed in cultured cortical neurons transfected with EGFP and found to be reduced in neurons lacking *BC1* RNA (Figs. 1a, b). Dendritic length was also decreased by ~ 20% in *BC1* KO neurons as compared to wild type (WT) neurons (Fig. 1c). Next, we analyzed spine density in EGFP-transfected neurons from both genotypes. Remarkably, *BC1* KO neurons have ~ 50% more spines than WT neurons (Fig. 1d). Thus, absence of *BC1* RNA causes robust effects on spine density and dendritic complexity, resembling alterations observed in neurons lacking FMRP[24, 28] or with decreased levels of CYFIP1[23].

We next used the barrel cortex, a well-established model to study neuronal connectivity and plasticity[22], to investigate the

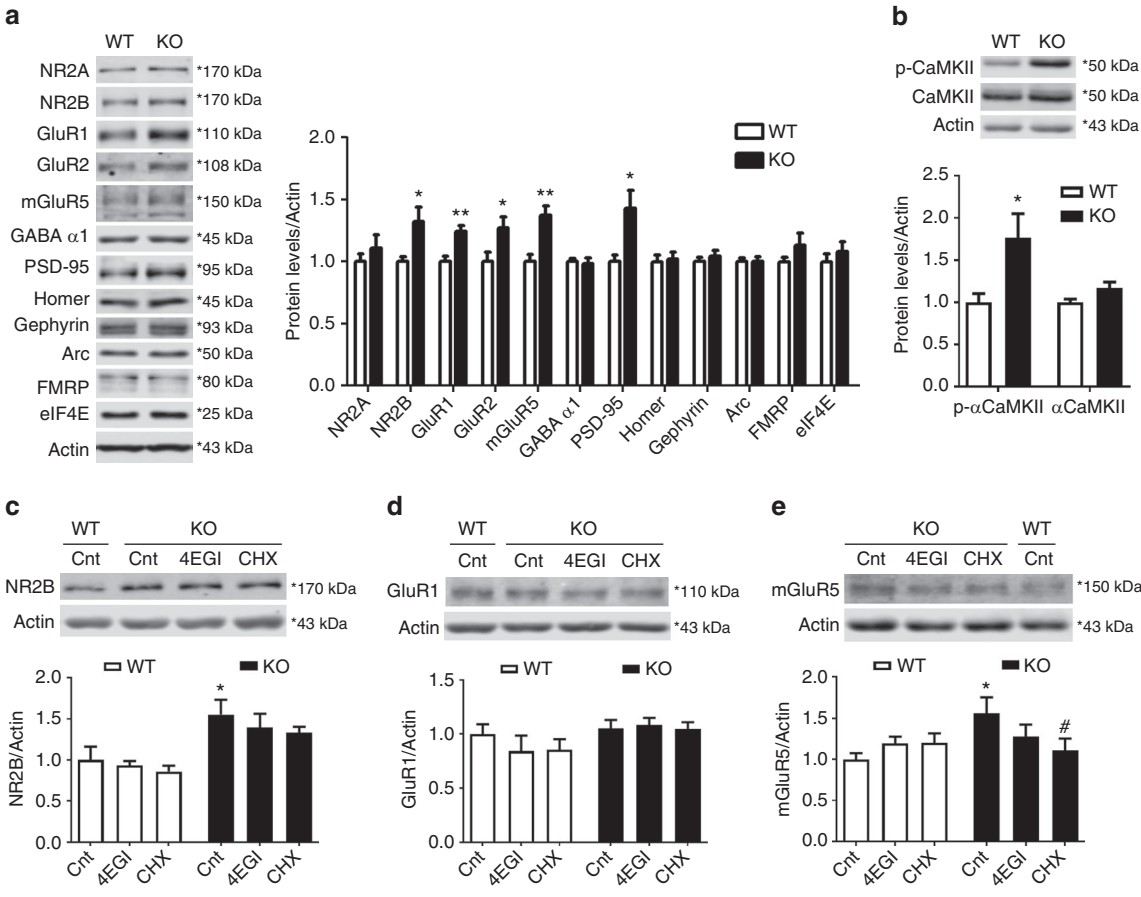

**Fig. 4** *BC1* KO mice show increased synaptic levels of GluRs and PSD-95. **a** *left* Representative western blotting of different GluR subunits and synaptic proteins in PSD-enriched fractions from WT and *BC1* KO mice. *right* Quantification of protein levels; values are expressed as ratio (fold of WT) of protein levels over actin (*$P < 0.05$, **$P < 0.01$, two-tailed $t$-test, mean ± s.e.m., $n = 6$ WT and 9 KO mice). **b** *top* Representative immunoblots of phospho-αCaMKII (Thr286) and total αCaMKII in PSD-enriched fractions from WT and *BC1* KO mice. *bottom* Quantification of protein levels; values are expressed as ratio (fold of WT) of protein levels over actin (*$P < 0.05$, two-tailed $t$-test, mean ± s.e.m., $n = 6$ WT and 9 KO mice). **c–e** *top* Representative western blotting of the indicated GluRs in WT and *BC1* KO synaptoneurosomes following 1 h treatment with vehicle (Cnt), 4EGI (50 μM), or cycloheximide (CHX, 25 μM). *bottom* Quantification of GluR over actin levels (*$P < 0.05$ vs. WT Cnt, #$P < 0.05$ vs. KO Cnt, two-way ANOVA, mean ± s.e.m., $n = 6$–13 mice). **c** WT Cnt ($n = 9$), WT 4EGI ($n = 7$), WT CHX ($n = 7$), KO Cnt ($n = 8$), KO 4EGI ($n = 8$), KO CHX ($n = 8$). **d** WT Cnt ($n = 13$), WT 4EGI ($n = 6$), WT CHX ($n = 6$), KO Cnt ($n = 11$), KO 4EGI ($n = 11$), KO CHX ($n = 11$). **e** WT Cnt ($n = 10$), WT 4EGI ($n = 7$), WT CHX ($n = 7$), KO Cnt ($n = 7$), KO 4EGI ($n = 7$), KO CHX ($n = 6$)

role of *BC1* RNA in vivo. In situ hybridization on brain sections containing the barrel region revealed that *BC1* RNA is expressed in the nucleus and cytoplasm of barrel cortex neurons (Supplementary Fig. 1). Neurons from young adult mice, after the critical period of circuit formation and early developmental plasticity[22], were labeled using Golgi staining and subdivided into pyramidal neurons from layers 2/3 and 5 and stellate cells from layer 4 in the barrel region of the somatosensory cortex. Both apical and basal dendrites were analyzed (Fig. 1e). *BC1* KO mice displayed significantly higher spine density than WT mice in principal neurons from layers 2/3 (Fig. 1f) and in stellate neurons from layer 4 (Fig. 1g). In contrast, no genotype differences were observed in layer 5 pyramidal neurons (Fig. 1h).

**BC1 RNA regulates spine morphology and PSD size.** Next, we analyzed spine morphology in layer 2/3 barrel cortex neurons. Remarkably, *BC1* KO mice have abnormally large spine heads as compared to WT mice (Figs. 2a, b; WT: median 0.701 ± 0.27 μm, 25/75% percentiles, 0.431/0.917 μm, $n = 1303$; *BC1* KO: median 0.836 ± 0.27 μm, 25/75% percentiles, 0.539/0.971 μm, $n = 2747$). Given the bimodal distribution of spine head size (with peaks at 0.4 and 0.8 μm for WT and *BC1* KO mice, respectively; Fig. 2b),

spine heads were classified as either large or small (see Methods). *BC1* KO neurons showed a significantly higher proportion of spines with large head as compared to WT mice (Fig. 2b). Spine morphology was further analyzed using transmission electron microscopy (TEM) from high-pressure frozen sections containing the barrel cortex (Fig. 2c). Frequency distribution of the observed PSD length (Fig. 2d, see Methods) revealed that *BC1* KO mice possess more spines with long PSD (24.9 vs. 31.1% for WT and *BC1* KO mice, respectively; $\chi^2 = 4.22$, *$P < 0.05$), whereas WT animals have a greater proportion of spines with short PSD (24.9 vs. 17.4% for WT and *BC1* KO mice, respectively; $\chi^2 = 8.11$, **$P < 0.01$). Furthermore, the PSD was significantly longer in *BC1* KO mice (324.9 ± 4.5 nm) than in WT mice (307.7 ± 5.5 nm) (*$P < 0.05$, Mann–Whitney $U$-test). We also performed a staining with phosphotungstic acid, which allows a better visualization of the PSD with TEM (Fig. 2e). Consistent with the data obtained with high-pressure frozen preparations, the PSD length was significantly increased in *BC1* KO mice as compared to their WT littermates (Fig. 2f). Likewise, the thickness of the PSD was also enhanced in mice lacking *BC1* RNA (Fig. 2g). In addition, there was a modest but significant increase in the length of the active zone (255.5 ± 11.3 and 287.8 ± 9.8 nm; $n = 99$ and 83 for WT and *BC1* KO mice, respectively; **$P < 0.01$, Mann–Whitney $U$-test).

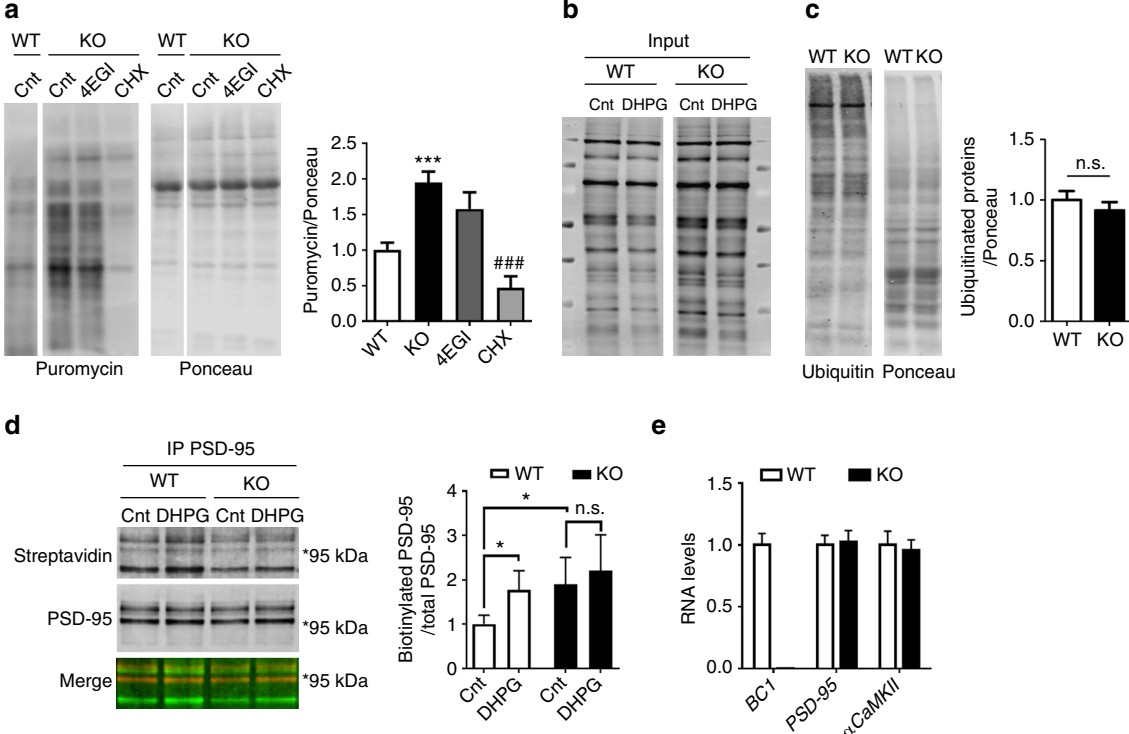

**Fig. 5** Translation of *PSD-95* mRNA is dysregulated in *BC1* KO synaptoneurosomes. **a** *left* Representative western blotting showing incorporated puromycin in WT and *BC1* KO cortical neurons. *right* Quantification of puromycin immunodetection normalized to Ponceau red staining (***$P < 0.001$ vs. WT, ###$P < 0.001$ vs. KO, one-way ANOVA, mean ± s.e.m., $n = 9$ WT, 16 KO (Cnt), 6 KO (4EGI) and 8 KO (CHX) mice). **b** Synaptoneurosomes from WT and *BC1* KO mice were incubated with vehicle (Cnt) or 100 μM DHPG for 10 min in presence of L-AHA. Representative western blotting for all de novo-synthesized (biotin-positive) proteins detected using IRDye 800CW Streptavidin. Blots are different parts of the same gel (see Supplementary Fig. 6 for full scans). **c** *left* Representative western blotting showing the levels of ubiquitinated proteins in WT and KO synaptoneurosomes. *right* Quantification of ubiquitinated proteins normalized to Ponceau red staining ($P = 0.4202$; two-tailed *t*-test, mean ± s.e.m., $n = 6$ WT and 7 KO mice). n.s., not significant. **d** *left* Western blotting for de novo-synthesized (biotinylated) and total PSD-95, after immunoprecipitation (IP) with PSD-95 antibody. *right* Quantification of immunoblots; data are presented as the ratio (fold of WT-control) of biotinylated PSD-95 over total PSD-95 (*$P < 0.05$, two-way ANOVA, mean ± s.e.m., $n = 6$ WT and 5 KO mice). **e** Levels of *BC1* RNA and of *PSD-95* and *αCaMKII* mRNAs in cortical extracts from WT (white bars) and *BC1* KO (black bars) mice. Values were normalized for *HPRT* and *Gusb* mRNA levels and expressed as fold of WT (mean ± s.e.m., $n = 5$ WT and 3 KO mice)

Furthermore, serial block face scanning electron microscopy (SBF-SEM) imaging followed by three-dimensional reconstruction of barrel cortex synapses confirmed the presence of abnormally large spine heads and PSDs in *BC1* KO mice (Fig. 2h and Supplementary Movies 1 and 2).

**BC1 KO pyramidal neurons have increased spontaneous activity.** To investigate whether the observed alterations in spine number and morphology correlate with concomitant changes in synaptic function, spontaneous excitatory postsynaptic currents (sEPSCs) were measured from layer 2/3 pyramidal neurons of the barrel cortex. Biocytin-labeled cells were recorded simultaneously (Fig. 3a). No changes were found in basic active or passive electrophysiological properties such as resting membrane potential, action potential threshold/half-width or input resistance between WT and *BC1* KO mice (Supplementary Fig. 2). The amplitude of sEPSC was significantly enhanced in *BC1* KO mice as compared to WT age-matched littermates (Fig. 3b). In contrast, no differences were found in sEPSC frequency (Fig. 3c). sEPSC decayed faster in *BC1* KO neurons than in WT neurons, but the difference was not significant (Fig. 3d). The increased sEPSC amplitude observed ex vivo could result in more cells reaching spiking threshold, thereby leading to enhanced cortical activity. In vivo juxtasomal recordings were therefore performed from layer 2/3 barrel cortex neurons (Fig. 3e). Pyramidal neurons from *BC1* KO mice showed a higher average spontaneous spiking

activity (Fig. 3f). The variability in spiking frequency was quantified by calculating the coefficient of variation (CV), which was also notably higher in *BC1* KO mice (CV = 59 and 109% for WT and *BC1* KO mice, respectively). Overall, our findings suggest that the increased in vivo spontaneous activity in *BC1* KO mice could result from sub-threshold changes of sEPSC (Fig. 3b).

Because alterations in spontaneous activity can affect cortical processing of tactile information[29, 30], we investigated stimulation-evoked cortical responses. Facial whiskers provide input to the mouse somatosensory barrel cortex[22]. Thus, one whisker at a time was deflected using a piezo wafer while recording the activity in the corresponding or adjacent barrel column (Fig. 3e). There was no significant difference in the average response to principal whisker (PW) stimulation (Fig. 3g). We found, however, that the variability in the average response to PW stimulation was higher in *BC1* KO mice as compared to WT mice (CV = 41 and 73% for WT and *BC1* KO mice, respectively). We also analyzed the cortical response when stimulating the first-order surrounding whiskers (SW). Similarly to PW stimulation, no significant differences were found in the average response rate to SW stimulation between genotypes (Fig. 3h), but the variability of the average response to SW stimulation was higher in *BC1* KO mice (CV = 59 and 129% for WT and *BC1* KO mice, respectively). In summary, the changes in synaptic properties caused by the absence of *BC1* RNA observed ex vivo, were manifested as an increase in the variability of the cortical activity recorded in vivo.

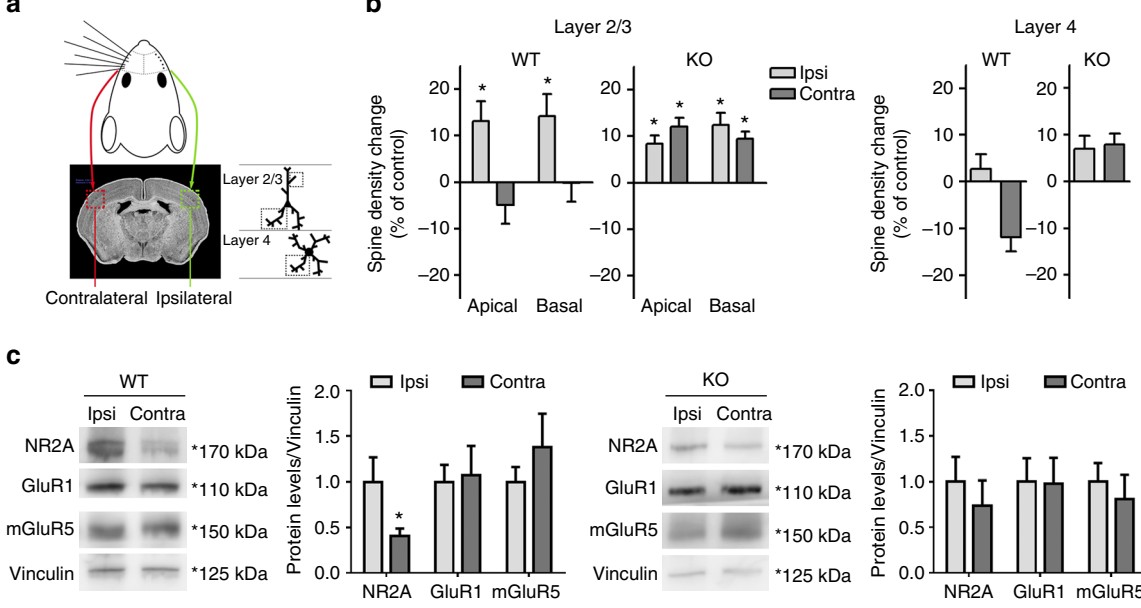

**Fig. 6** *BC1* KO mice show abnormal spine plasticity in response to whisker deprivation. **a** Schematic illustrating the cortical layers and dendrites of the somatosensory cortex analyzed 1 week after whisker deprivation. The brain slice photograph was taken from MBL brain atlas of the C57BL/6j mouse[82] (http://www.mbl.org/atlas170/atlas170_frame.html). **b** Changes in spine density after whisker deprivation in WT and *BC1* KO mice were detected by Golgi staining and are expressed as percentage relative to their respective control groups (*$P < 0.05$ as compared to control, Mann–Whitney *U*-test, mean ± s.e.m., layer 2/3: $n = 15/14$ and 14/13 neurons for I/C WT and KO mice, respectively; layer 4: $n = 12$ and 13 neurons for WT and *BC1* KO mice, respectively). **c** Representative immunoblots (*left*) and quantitative analysis (*right*) of the levels of GluR subunits in the ipsilateral [Ipsi] and contralateral [Contra] barrel cortex of WT and *BC1* KO mice 1 week after whisker deprivation. Values expressed as ratio (fold of ipsilateral) of GluR over Vinculin protein levels (*$P < 0.05$, two-tailed *t*-test, mean ± s.e.m., $n = 9$ WT and 7 KO mice)

The increased sEPSC amplitude (Fig. 3b) and spontaneous activity (Fig. 3f) suggest an overall increase in cortical neuronal activity in *BC1* KO mice. To further investigate this observation, background metabolic activity was assessed using cytochrome oxidase (CO) histochemistry. Indeed, CO staining was significantly increased in the barrel cortex of *BC1* KO mice as compared to WT mice (Fig. 3i), indicating that *BC1* KO barrel cortex neurons have a higher basal metabolic rate.

**BC1 KO mice have increased glutamate receptors levels.** Lack of *BC1* RNA resulted in enhanced sEPSC in the somatosensory cortex (Fig. 3). However, whether this reflects overall changes in glutamate receptors (GluRs) expression or alteration in their synaptic localization is currently unknown. We then analyzed the postsynaptic expression of different GluR subunits and associated scaffold proteins in PSD-enriched preparations from the cortex of WT and *BC1* KO mice (Supplementary Fig. 3a, see Methods). Expression of both GluR1 and GluR2 was increased in postsynaptic membranes from *BC1* KO mice (Fig. 4a). Likewise, mGluR5 and NR2B, but not NR2A, levels were higher in *BC1* KO mice as compared to WT mice. Of note, expression of these GluR subunits remained unchanged in whole-cortex lysates from *BC1* KO mice as compared to WT mice (Supplementary Fig. 3b). Interestingly, the levels of PSD-95 were also significantly enhanced in PSD-enriched fractions from *BC1* KO mice, whereas Homer, Gephyrin or the GABA$_A$ receptor subunit α1 levels did not. Other proteins that regulate GluR trafficking or mRNA translation (i.e., Arc, FMRP and eIF4E) did not change in the PSD-enriched preparations (Fig. 4a). A biotinylation assay in primary cortical neurons confirmed that levels of membrane-associated NR2B, GluR1 and GluR2 are higher in *BC1* KO compared to WT neurons (Supplementary Fig. 3c). Because αCaMKII has been implicated in GluR synaptic delivery and it is also an FMRP target regulated by *BC1* RNA in intact

synapses[11, 17], we examined αCaMKII levels in postsynaptic membrane preparations from WT and *BC1* KO mice. While total expression of αCaMKII was slightly increased, the levels of phosphorylated αCaMKII (pT286) were markedly enhanced in PSD-enriched preparations from *BC1* KO mice as compared to WT mice (Fig. 4b), indicating that *BC1* KO mice have higher levels of active, membrane-associated αCaMKII at synapses.

The observed alterations of postsynaptic proteins in *BC1* KO mice could be associated with modifications of the actin cytoskeleton involving membrane receptor trafficking[31]. However, the relative amount of filamentous actin (F-actin) vs. globular actin (G-actin) in cortical lysates from WT and *BC1* KO mice was comparable (Supplementary Fig. 3d).

*BC1* RNA increases the affinity of the translational repressor FMRP for its target mRNAs[11, 16]. *GluR1* and *NR2B* are known FMRP target mRNAs[32, 33], we therefore analyzed GluR expression levels in synaptoneurosomes following acute treatment with protein synthesis inhibitors. NR2B levels were increased in *BC1* KO synaptoneurosomes as compared to WT; the effects were not affected by neither a specific inhibitor of cap-dependent translation (4EGI) nor the general translation inhibitor cycloheximide (Fig. 4c), possibly due to the slow turnover rate of the receptor[34]. In contrast, GluR1 expression was not different between genotypes and remained unchanged under all experimental conditions (Fig. 4d). The levels of mGluR5 in *BC1* KO synaptoneurosomes were elevated compared to WT mice, and while 4EGI only partially reduced mGluR5 expression in *BC1* KO synaptoneurosomes, cycloheximide completely reversed it (Fig. 4e). None of the inhibitors modified basal mGluR5 expression in WT synaptoneurosomes (Fig. 4e). Considering that the expression of GluRs in whole-cortex lysates was not different between genotypes (Supplementary Fig. 3), these results suggest that *BC1* RNA regulates dendritic translation of *mGluR5* and *NR2B* but not of *GluR1* mRNA. To gain insights into the possible

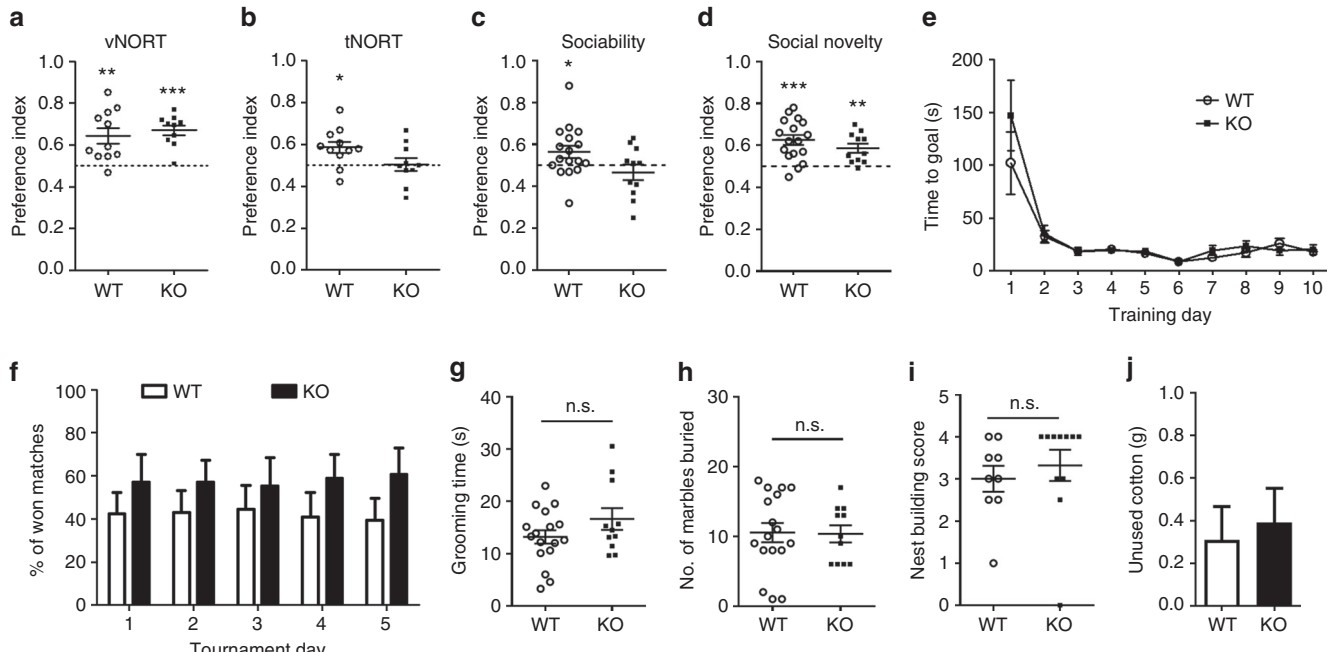

**Fig. 7** *BC1* KO mice have impaired texture novel object recognition (tNORT) and social behavior. **a**, **b** Preference index (novel vs. familiar object) in the visual novel object recognition test (vNORT) and tNORT (*$P < 0.05$, **$P < 0.01$, ***$P < 0.001$ vs. chance level, one-sample *t*-test, mean ± s.e.m., $n = 11$ WT and 10 KO mice). **c**, **d** Preference index for sociability (stranger 1 vs. empty cage) and for social novelty (stranger 2 vs. stranger 1) in the three-chamber test (*$P < 0.05$, **$P < 0.01$, ***$P < 0.001$ vs. chance level, one-sample *t*-test, mean ± s.e.m., $n = 17$ WT and 11 KO mice). **e**, **f** Time to reach the goal during the training sessions and percentage of matches won in the automated tube test ($n = 8$ mice). **g**, **h** Time spent self-grooming and number of marbles buried by WT and *BC1* KO mice ($P = 0.1516$ and $0.9336$, respectively, two-tailed *t*-test, mean ± s.e.m., $n = 17$ WT and 11 KO mice). **i**, **j** Nest-building test score and unused cotton material ($P = 0.5313$ and $0.7289$, respectively, two-tailed *t*-test, mean ± s.e.m., $n = 9$ WT and 11 KO mice). n.s., not significant

molecular mechanism, we looked for potential complementary regions within the *BC1* RNA sequence. Similarly to other FMRP target mRNAs partially regulated by the FMRP-*BC1* complex through sequence complementarity[11], *NR2B* and *mGluR5* mRNAs show a higher putative complementarity with *BC1* RNA compared to *GluR1* mRNA (Supplementary Fig. 4). The bioinformatics predictions are consistent with our experimental findings in synaptoneurosomes, and suggest that *BC1* RNA might directly regulate the translation of some but not all GluR subunits.

**Dysregulated activity-dependent translation in *BC1* KO mice**. We next tested whether the absence of *BC1* RNA affects translational activity in neurons by using the SUnSET assay[35]. A twofold increase in global protein translation was found in *BC1* KO neurons compared to WT neurons (Fig. 5a). The cap-dependent translation inhibitor 4EGI partially attenuated the increase in puromycin incorporation in *BC1* KO neurons but the effect did not reach statistical significance, while treatment with the nonspecific translation inhibitor cycloheximide completely inhibited protein synthesis (Fig. 5a). We further investigated whether *BC1* RNA regulates local protein translation at synapses via metabolic labeling of cortical synaptoneurosomes. In brief, a modified aminoacid, L-azidoalanine (L-AHA), was incorporated into nascent proteins in basal and DHPG-stimulated conditions and subsequently biotinylated using the Click-It technology[36]. An overall increase in L-AHA incorporation was found in *BC1* KO mice as compared to WT mice (Fig. 5b).

The increase in basal metabolic activity (Fig. 3) and protein translation (Fig. 5) observed in *BC1* KO mice could be associated with enhanced protein turnover. To explore this possibility, we examined the levels of ubiquitinated proteins in synaptoneurosomes under steady state. However, no differences were found

between WT and *BC1* KO mice (Fig. 5c). These results indicate that there is an unbalance between protein synthesis and degradation at *BC1* KO synapses, a phenomenon that has been associated with deficits in synaptic plasticity as well as with ASD[37].

Because *BC1* KO mice have larger spine heads and PSD size (Fig. 2) as well as increased PSD-95 levels (Fig. 4), we investigated whether *BC1* RNA participates in the regulation of local PSD-95 synthesis. Metabolically-labeled synaptoneurosomes (Fig. 5b) were immunoprecipitated with a PSD-95 antibody in order to detect *de novo* translated PSD-95. An L-AHA-labeled band was detected at 95 kDa overlapping with the signal obtained with PSD-95 antibody (Fig. 5d), which confirmed the identity of the band as newly synthesized PSD-95. The rate of PSD-95 translation in WT synaptoneurosomes, relative to total immunoprecipitated PSD-95, was increased after stimulation with the mGluR1/5 agonist DHPG. In *BC1* KO synaptoneurosomes, the basal rate of PSD-95 translation was enhanced compared to WT mice, and DHPG had no further effect (Fig. 5d). Importantly, the mRNA levels encoding PSD-95 and αCaMKII remain unchanged in cortical extracts from *BC1* KO mice as compared to WT mice, excluding an effect of *BC1* RNA on transcription or mRNA degradation (Fig. 5e). Taken together, these results indicate that *BC1* RNA regulates PSD-95 synthesis at cortical synapses. *PSD-95* mRNA shares a nucleotide complementary with *BC1* RNA (Supplementary Fig. 4), which supports a possible direct regulation of *PSD-95* mRNA translation mediated by *BC1* RNA.

**_BC1_ KO mice show aberrant experience-dependent plasticity**. Next, we investigated the role of *BC1* RNA on experience-dependent plasticity in the somatosensory cortex. Sensory

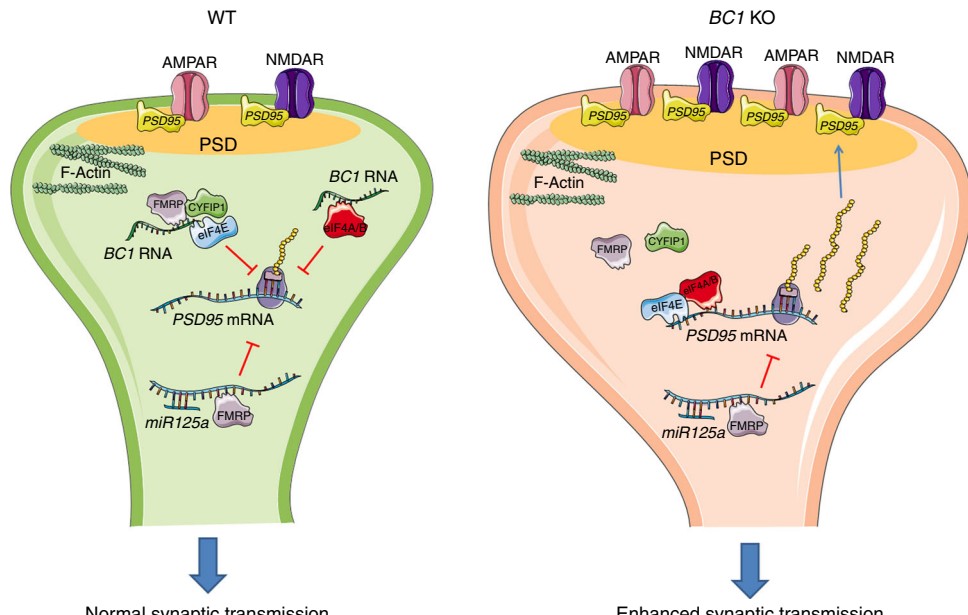

**Fig. 8** *BC1* RNA-mediated regulation of structural plasticity at cortical synapses. In WT mice, *PSD-95* mRNA translation is repressed through multiple mechanisms including regulation by FMRP and *miR125a*[65], association of *BC1* RNA with the FMRP-CYFIP1 complex[12, 17], and interaction between *BC1* RNA and eIF4A/B[15, 16]. These different ribonucleoprotein complexes mediate activity-regulated translational control of *PSD-95* mRNA. Absence of *BC1* RNA causes exaggerated local synthesis of PSD-95 and enhanced synaptic delivery of ionotropic glutamate receptors to the postsynaptic density (PSD). These biochemical alterations along with the oversized PSD and spine heads might contribute to the increase in synaptic transmission and to the behavioral abnormalities observed in *BC1* KO mice. The figure was produced using Servier Medical Art (http://www.servier.com)

deprivation by whisker trimming has been consistently shown to change neuronal connectivity in the rodent barrel cortex, having opposite effects in the contralateral and ipsilateral hemispheres relative to the trimmed whiskers[38–40]. This plastic response is accompanied by changes in spine number of barrel cortex neurons[41, 42]. We therefore analyzed spine density in neurons of layers 2/3 and 4 from both the ipsilateral and contralateral barrel cortex 1 week after whisker deprivation (Fig. 6a). As reported previously[41, 42], sensory deprivation altered spine density in both apical and basal dendrites of layer 2/3 pyramidal neurons and layer 4 spiny stellate cells in rodents; interestingly, here we find that in deprived WT mice spine density of layer 2/3 neurons was significantly increased in the ipsilateral side, while layer 4 neurons showed reduced number of spines in the contralateral but not the ipsilateral hemisphere, as compared to non-deprived (control) mice (Fig. 6b). In marked contrast, unilateral whisker trimming in *BC1* KO mice led to an overall increase in spine density in layer 2/3 and 4 neurons from both the ipsilateral and contralateral barrel as compared to non-deprived control *BC1* KO mice. These findings indicate that structural plasticity in response to sensory deprivation is abnormal in *BC1* KO mice.

Whisker deprivation has been associated with changes in synaptic levels of GluRs[43]. We therefore determined the levels of different GluR subunits in the somatosensory cortex of sensory-deprived mice. In WT mice, a significant decrease in NR2A but not GluR1 or mGluR5 was detected in the contralateral cortex as compared to the ipsilateral one (Fig. 6c). In contrast, the levels of these receptors were not statistically different between hemispheres in *BC1* KO mice (Fig. 6c). NR2A expression in the somatosensory cortex was not statistically different between WT and *BC1* KO non-deprived mice ($P = 0.4236$, two-tailed *t*-test, $1.00 \pm 0.237$ and $0.742 \pm 0.199$ for WT and *BC1* KO mice, respectively, mean ± s.e.m., $n = 6$ mice). Thus, alterations in spine density in response to sensory deprivation seem to correlate with changes in NMDA receptor expression in the somatosensory cortex.

**BC1 KO mice show deficits in learning and social behavior**. The novel object recognition test (NORT) is a well-established memory task that involves the processing of sensory information from whiskers to the barrel cortex[44, 45]. Two different paradigms were used, visual NORT (vNORT) and texture NORT (tNORT)[46], to assess the contribution of visual and whisker-dependent sensory experience to learning, respectively. In addition, a short time delay of 5 min was used between habituation and test to limit the contribution of hippocampus and maximize the contribution of cortical areas to the task[47]. Both WT and *BC1* KO mice had preference for novel over familiar object in the vNORT paradigm (Fig. 7a, and Supplementary Fig. 5a). In the tNORT paradigm; however, only WT mice showed an overall preference for novel vs. familiar object (Fig. 7b). Importantly, there were no differences in total exploration time between genotypes in tNORT (Supplementary Fig. 5b). Taken together, our findings suggest that deficits in tNORT in *BC1* KO mice are not related to lack of novelty-seeking behavior, as they performed similarly to WT mice in the vNORT, but are likely due to impaired processing of somatosensory information.

Reduced social interaction and repetitive behaviors are key features of neuropsychiatric disorders, including autism and FXS[48, 49]. Moreover, social behavior is affected in mice with reduced thalamocortical input into the barrel cortex[50]. We examined whether *BC1* KO mice present alterations in social interaction and preference for social novelty using the three-chamber paradigm. Interestingly, *BC1* KO mice showed defects in sociability but had intact social memory (Figs. 7c, d and Supplementary Figs. 5c, d). These results further support the idea that novelty-seeking behavior is intact in *BC1* KO mice. In the automated tube test, a social dominance paradigm, both WT and *BC1* KO mice showed a normal learning curve during the training session, as measured by the time spent to reach the goal (Fig. 7e). No differences in the number of matches won per tournament were found between genotypes (Fig. 7f), indicating normal social hierarchic behavior in *BC1* KO mice. We next

evaluated self-grooming and marble burying, two behavioral tests commonly used to assess stereotypic and compulsive behaviors and anxiety, respectively[48]. On average, *BC1* KO mice spent more time self-grooming as compared to WT mice but this difference was not statistically significant (Fig. 7g). Likewise, no significant differences were found between WT and *BC1* KO mice in the number of buried marbles (Fig. 7h).

We also analyzed nest building performance in *BC1* KO mice, a natural home-cage behavior that is affected in mouse models of FXS and ASD[48, 51]. However, no differences were found between genotypes as measured by scoring the quality of the nest and the amount of unused cotton material (Figs. 7i, j). In summary, our results indicate that *BC1* KO mice have learning and social deficits, behavioral phenotypes commonly observed in animal models of ASD, including FXS[48, 49] (Supplementary Table 1).

## Discussion

Perturbations in ncRNAs, including *BC1* and its human orthologue *BC200*, are associated with a number of neurological diseases[2, 3]. *BC1* functions in part through its association with FMRP[11], the protein absent or mutated in FXS, which is the major hereditary cause of intellectual disability and also the most prevalent monogenic factor of ASD. Here we show that the absence of *BC1* RNA leads to alterations in spine density and morphology in the rodent barrel cortex, increased postsynaptic levels of GluRs, and enhanced spontaneous neuronal activity ex vivo and in vivo. Furthermore, we provide evidence that experience-dependent plasticity and learning are impaired in *BC1* KO mice.

Defects in the regulation of activity-dependent spine formation and maturation are common features associated with mutations in genes implicated in intellectual disabilities and ASD[25, 28, 52]. Significant alterations in spine number and morphology have been extensively described in mouse models for neurodevelopmental disorders[25, 52]. Absence of *BC1* RNA resulted in increased spine density and decreased dendritic complexity, similar to mice deficient for FMRP[24, 28] or its binding partner CYFIP1[23, 53]. Conversely, *BC1* KO mice had larger spine heads than WT mice, as opposed to what has been observed in mouse models of FXS and ASD[25, 53], but similar to observations in animal models of Down Syndrome[54]. We hypothesize that dendritic elaboration and spine formation may be affected by the absence of the FMRP-CYFIP1-*BC1* complex, whereas spine head and PSD size could be under the control of alternative *BC1*-containing complex/es. In support of this hypothesis, *BC1*-mediated regulation of protein synthesis, neurotransmission and behavior can occur via FMRP-dependent or FMRP-independent mechanisms[5, 9, 11, 15–17, 20, 21, 55, 56]. Interestingly, overexpression of PSD-95 increases the number and size of spines[57] to a similar extent as observed here in mice lacking *BC1* RNA. Spine growth and stabilization require activity-dependent PSD-95 expression and local translational activity[57–59]. In the *BC1* KO mice, the observed elevated PSD-95 levels in PSD-enriched fractions and the loss of activity-regulated PSD-95 synthesis in synaptoneurosomes could therefore contribute to the observed spine abnormalities. Likewise, the increase in phosphorylated αCaMKII at *BC1* KO synapses could also be involved in their spine phenotype, as the overexpression of constitutively active αCaMKII increases both spine size and density[60].

*PSD-95* mRNA is a well-characterized FMRP target that is rapidly translated after synaptic stimulation[32, 61–63]. The underlying mechanisms and regulatory proteins implicated in *PSD-95* mRNA translation appear to differ between brain regions; while FMRP modulates *PSD-95* mRNA stability in the hippocampus, it represses PSD-95 translation at cortical synapses (for review[64]),

the latter probably involving CYFIP1 and *BC1* RNA[16] as well as the microRNA 125a[61] (Fig. 8). The interaction between CYFIP1 and FMRP is stabilized by *BC1* RNA, an important event for translational regulation[11, 16]. Furthermore, *BC1* associates with FMRP-containing RNPs which translocate to spines in response to mGluR1/5 activation[10]. Our findings suggest that *BC1* RNA participates in FMRP-mediated translational control of PSD-95 and possibly other GluRs. Thus, we propose a model in which absence of *BC1* RNA releases the inhibition of protein translation, and as a result PSD-95 and specific GluR subunits are locally synthesized at high rates, leading to exaggerated synaptic targeting of GluRs (Fig. 8). This, in turn, would cause augmented sEPSC as well as enlargement of PSDs and spine heads, which could account for the enhanced spontaneous activity and response variability to whisker stimulation found in vivo. This sequence of events is consistent with the notion that changes in spine morphology are not intrinsic to certain neurological diseases only but may be a consequence of alterations in excitatory synaptic input[25].

Previous studies have reported abnormal mGluR-dependent transmission in the striatum and hippocampus of *BC1* KO mice[4, 21]. However, whether these alterations result from changes in mGluR expression and/or membrane localization or enhanced receptor activity has not yet been elucidated. Here, we found that *BC1* KO mice have increased mGluR5 expression in postsynaptic membranes, suggesting that the upregulation of type-I mGluR-mediated transmission[4, 21] might be due to changes in synaptic mGluR5 levels. In addition, membrane-associated expression of NR2B, GluR1 and GluR2 were also increased in *BC1* KO mice in vitro and ex vivo conditions, a finding consistent with the observed enhancement of sEPSC. Previous studies have shown that mRNAs encoding for GluR1, GluR2, and NR2B are FMRP targets[32, 33]. Because the overall GluR1 expression in cortical lysates and synaptoneurosomes remained unchanged between WT and *BC1* KO mice, *BC1* RNA likely regulates GluR1 trafficking to the postsynaptic membrane rather than its synthesis. In this regard, the increase in PSD-95 levels, a well-known scaffolding protein of ionotropic GluR, together with the enhanced αCaMKII phosphorylation at synapses (which is known to regulate GluR trafficking) might be sufficient to drive GluR delivery to the PSD[57, 65–67]. In contrast, our data indicated that expression of NR2B and mGluR5 was enhanced in *BC1* KO synaptoneurosomes, an effect reversed by protein synthesis inhibitors in the case of mGluR5. Furthermore, the sequence complementarity prediction between *BC1* RNA and *NR2B* and *mGluR5* mRNAs supports the involvement of this ncRNA in local translation of these GluR subunits.

Spine morphology is heavily influenced by changes in neuronal activity via synaptic GluRs[68]. Our findings are in agreement with this model, showing that there is a correlation between the spine phenotype, synaptic GluR levels and glutamatergic synaptic responses. This notion is further supported by our whisker trimming data in WT mice, where a decrease in spine density in the contralateral barrel cortex was associated with reduced NR2A levels as compared to the ipsilateral hemisphere. In line with this observation, previous studies have found similar changes in spine density and synaptic NMDA receptors in layers 2/3 and 4 of the somatosensory cortex after whisker deprivation[41–43]. Notably, NR2A expression in the cortex is strongly regulated by synaptic activity and sensory experience[69–72]. A previous study reported that whisker trimming attenuates spine loss in layer 5 pyramidal neurons compared to non-deprived mice[73], indicating a diverse regulation of structural plasticity in different neuronal populations.

Of note, a large-scale mass spectrometry study found accumulation of proteins and enzymes involved in protein degradation at deprived synapses of the barrel cortex[43], which could be

responsible for the decrease in NR2A expression observed here. The exaggerated synaptic levels of PSD-95 in *BC1* KO mice could prevent NMDA receptor degradation induced by whisker trimming, thereby resulting in enhanced spine density. Consistently, interaction with PSD-95 blocks NMDA receptor internalization[68], a necessary step for its degradation. Furthermore, reduced PSD-95 ubiquitination has been described to hamper activity-dependent synapse elimination in neurons lacking FMRP[74], an effect that could also contribute to the increase in spine density and NR2A levels observed in *BC1* KO mice following sensory deprivation. Interestingly, similar alterations in spine turnover have been found in the barrel cortex of FXS mice after unilateral whisker trimming[73], suggesting a common mechanism for FMRP and *BC1* RNA in the modulation of sensory experience-dependent spine dynamics.

*BC1* KO mice showed impaired texture recognition as well as deficits in sociability, behavioral abnormalities typically observed in rodent models of FXS and ASD[48, 49] (Supplementary Table 1). Importantly, the somatosensory cortex is required for optimal exploration and detection of a novel object[43, 45] and it also plays a role in social behavior[50]. Given that alterations in spontaneous neuronal activity can affect cortical processing of whisker-dependent tactile information[29, 30], we hypothesize that the increased spiking variability observed in the barrel cortex of *BC1* KO mice could contribute to deficiencies in texture recognition, a whisker-dependent discrimination task[75], as well as in sociability. Although sociability is a complex trait entailing several interconnected brain regions, previous studies have reported alterations in multiple neurotransmitter systems and increased neuronal hyper-excitability in the striatum[9, 20, 21] and hippocampus[4] of *BC1* KO mice, which may underlie their behavioral phenotype. Furthermore, the alterations in spine morphology and cortical plasticity described here could contribute to the reduced exploratory behavior and survival rate of *BC1* KO mice reported in a natural, large-scale open field[6], especially considering the importance of the barrel cortex in processing sensory information[22]. In summary, our results indicate that *BC1* RNA plays a crucial role in fine-tuning the regulation of synaptic structure and function, and it is required for sensory experience-dependent plasticity and learning.

## Methods

**Animal care**. Animal care was conducted in accordance with the Belgian, Swiss, and European laws, guidelines and policies for animal experimentation, housing, and care (Belgian Royal Decree of 29 May 2013 and European Directive 2010/63/EU on the protection of animals used for scientific purposes of 20 October 2010). All the experimental procedures performed in Switzerland complied with the Swiss National Institutional Guidelines on Animal Experimentation and were approved by the Cantonal Veterinary Office Committee for Animal Experimentation. In all cases special attention was given to the implementation of the 3 R's, housing and environmental conditions and analgesia to improve the animals' welfare. Animals were housed with food and water ad libitum at 21 °C and with 12-h light/dark cycle. Six to 8 week-old WT and *BC1* KO[19] male mice (C57Bl/6 and 129Sv-C57Bl/6 background) were used for all the experiments, except for behavioral experiments (see below). *BC1* KO mice (129Sv-C57Bl/6) were backcrossed to the C57Bl/6 background for several generations, before they were used for experiments, and were bred in heterozygosity to provide WT and *BC1* KO littermates. The genetic background of the strains (WT and *BC1* KO) used throughout this study was also evaluated in a screening performed by Charles River (Charles River Genetic Testing Services, Wilmington, MA USA).

**Slice preparation and electrophysiology**. Brains from 7-week-old *BC1* KO and WT littermates were rapidly removed and dissected in ice-cold sucrose solution containing the following (in mM): 212.5 sucrose, 3.5 KCl, 1.2 KH$_2$PO$_4$, 3 MgSO$_4$, 1 CaCl$_2$, 26 NaHCO$_3$ and 10 glucose. Thalamocortical barrel cortex slices (300–400 μm) containing the barrel subfield of somatosensory cortex were cut on a vibrating microtome and then placed in a submerged-style holding chamber in artificial cerebrospinal fluid, bubbled with carbogen (95% O$_2$, 5% CO$_2$), containing the following (in mM): 125 NaCl; 3 KCl; 1.25 NaH$_2$PO$_4$; 1 MgSO$_4$; 2 CaCl$_2$; 26 NaHCO$_3$; 10 glucose. Slices were kept at 35 °C for 20 min following slicing before returning to room temperature for a further 40 min before being ready for recording.

Whole-cell patch-clamp recordings were made from pyramidal cells in superficial layers 2/3 of the somatosensory cortex at 32 °C. Neurons were identified under visual guidance by differential interference contrast microscopy and selected upon the basis of morphology and basic active/passive properties, e.g., spike half-width, input resistance, and resting membrane potential. Cell identity was re-confirmed and post-hoc spine analysis carried out in a selection of these cells that were filled with biocytin (0.2%) and processed further with chromogen 3,3′diaminobenzidine tetrahydrochloride (DAB) using the avidin–biotin–peroxidase method. Pipettes were made from standard borosilicate capillary glass tubing with resistance between 3.5 and 4.5 MOhm resistance and filled with intracellular solution containing (in mM): 140 Kgluconate; 9 KCl; 10 HEPES; 4 K$_2$phosphocreatine; 4 ATP-Mg; 0.4 GTP (pH 7.2–7.3, pH adjusted with KOH; 290–300 mOsm). Series resistance was not compensated for.

For recording sEPSCs, cells were held in voltage-clamp and recorded for between 4–12 min at −70 mV. GABAergic synaptic currents were not pharmacologically blocked but the intracellular chloride concentration was adjusted such that it did not differ significantly from the chloride reversal potential at the membrane holding potential so all downward current events were due to excitatory synaptic currents. All recordings were acquired at 10 kHz, filtered at 3 kHz, and acquired using Clampex 9.2 software. Basic cell properties and synaptic currents were analyzed in Igor Pro (Wavemetrics, OR, USA) and Mini Analysis (Synaptosoft, NJ, USA) with event detection levels for synaptic currents set at 8 pA.

**In vivo electrophysiology**. Each animal was pre-anesthetized with isoflurane. Next, an intraperitoneal injection of a Urethane/Acepromazine Maleate mix was delivered to anesthetize the animal for the electrophysiological procedures (70% of 1.0 g/kg dosage titrated based on animal weight). During the recordings, animal temperature was maintained and kept stable at 36.5–37.5 °C. Animal's condition was sustained by constantly providing oxygen, while the depth of anesthesia was checked regularly and supplemented by additional doses of anesthetics, as previously described[76]. In short, to maintain anesthesia at similar depth in all the recordings, breathing rate and spontaneous whisker movements were monitored along with hind leg withdrawal and corneal (blinking) reflexes that were checked every 15–30 min. Recordings were collected only in the medium range of anesthesia in order to avoid deep and light states of anesthesia. Furthermore, when supplementing anesthesia (10% of initial dosage), recordings were continued no earlier than 15 min after the injection. A piezo-electric stimulator was used to move single whiskers mechanically while simultaneously recording cortical responses (25 deflections at 1 Hz) with patch pipettes (resistance 4–8 MΩ). Juxtasomal recordings from layer 2/3 pyramidal cells were performed in the barrel column corresponding to the PW. The PW is defined as the whisker that elicits the highest number of spikes per stimulation. In addition to the PW, the spiking activity evoked when stimulating the first-order SW was also recorded. This was done both as a control to determine which whisker is PW and also to compare the whisker evoked response from the different whiskers. Whisker-stimulation-evoked responses were recorded from the same pyramidal cells that were used for the spontaneous activity recordings. Statistical comparisons between genotypes were done using unpaired *t*-test with Welch's correction, as variances were significantly different.

**Cytochrome oxidase (CO) histochemistry**. Animals were anesthetized with sodium pentobarbital (Nembutal, 600 mg/kg; Ceva Santa Animale, Kansas City, KS) by intraperitoneal injection before being sacrificed by cervical dislocation. The brains were rapidly removed and immediately frozen in 2-methylbutane (Merck, Overijse, Belgium) at −40 °C and stored at −80 °C until sectioning. Series of 25-μm-thick coronal sections were prepared on a cryostat (Microm HM 500 OM, Walldorf, Germany), mounted on 0.1% poly-L-lysine (Sigma-Aldrich, St. Louis, MO) coated slides and stored at −20 °C until further processing.

For CO reaction, slices were treated for 30′ at 37 °C with the following solution: 0.1 M Tris-HCl pH 7.6, 117 mM sucrose, 3.16 mM nickel hexahydrate, 3.16 mM ammonium sulfate, 16.15 μM cytochrome C and 2.78 mM DAB. Sections of WT and KO animals were stained in batch to guarantee identical staining conditions. Adjacent serial cryostat sections were Nissl counterstained (cresyl violet 1%, Fluka Chemika, Sigma-Aldrich) according to standard procedures. Images of the CO and Nissl-stained sections were obtained with a light microscope (Zeiss Axio Imager Z1) equipped with an AxioCam MRm camera (1388 × 1040 pixels) using the software program Zen 2012 (Blue Edition; Carl Zeiss-Benelux). Quantitative analysis of the optical density of the CO staining was accomplished by ImageJ software (NIH, Bethesda, Maryland, USA). For each animal (*n* = 3, per genotype), six barrels of at least two different coronal sections were analyzed.

**Golgi staining**. Golgi staining was performed using a previously described protocol[24]. Spine density was measured on pyramidal neurons located in layer 2/3 and layer 5 and on spiny stellate cells located in layer 4 of the barrel cortex of 7-week-old WT and *BC1*-KO mice. In some experiments, unilateral whisker trimming was performed in 6-week-old mice, and 1 week later spine density was analyzed. Neurons were first identified with a light microscope (Leica DMLB)

under low magnification (×20, NA 0.5) and were chosen by first locating, among all the stained sections, the regions of interest in their respective coronal sections. Four neurons in each layer showing at least fourth-order branches for both apical and basal dendrites were selected. Since no significant interhemispheric differences were observed in control WT or *BC1* KO mice, measurements were pooled, so that eight neurons per layer (24 neurons per brain) were studied in each animal. Only neurons which satisfied the following criteria were chosen for analysis in each of the experimental groups: (1) presence of untruncated dendrites; (2) all dendrites showed a consistent and dark impregnation along their entire extent; and (3) relative isolation from neighboring impregnated neurons to avoid interference and ensure accuracy of dendritic spine counting. Subsequently, dendritic spines were analyzed under a higher magnification (×63, NA 0.75). Series of sequential photomicrographs of dendritic segments were generated on a computer screen by means of a video camera connected to the microscope. These photomicrographs were acquired using at least five serial focal planes (2–3 μm apart) by focusing in and out with the fine adjustment of the microscope to create a stack of sequential images. This ensured the accurate reconstruction of entire dendritic segments and enabled counting of these segments with all their visible spines on two-dimensional (2D) images. To further minimize any missing of spines, both apical and basal dendrites of the selected segments had to be observed as much as possible in the series of focal planes selected for counting. Spines were counted on secondary and tertiary branches of apical (layer 2/3 and layer 5 cells) and basal (layer 2/3, layer 4 and layer 5 cells) dendrites. On each neuron and for each dendrite category, five 20 μm dendritic segments were randomly selected using a 2D sampling grid made of 20 μm squares to generate values of spine density. In some cases, segments were from the same branch. Segments were sampled 50 μm (apical dendrites) or 25 μm (basal dendrites) away from the soma in order to exclude the spine-depleted zone, which arises from the cell body. Only protuberances with a clear connection to the shaft of the dendrite were counted as spines. Since this method has proven to provide reliable results[77], no attempt was made to introduce a correction factor for hidden spines. As no difference in spine counts was observed between secondary and tertiary branch segments for each group, data were pooled for each dendrite category (basal and apical) to generate the final spine density results. An experimenter blind to the experimental conditions performed all measurements.

**Biocytin staining**. For a subset of recorded layer 2/3 pyramidal neurons, slice tissue was fixed in 4% paraformaldehyde and subsequently processed using DAB following the avidin–biotin–peroxidase method[78]. Dendrite sampling was carried out in an identical manner to that for Golgi-stained tissue. Briefly, consecutive stack images of randomly sampled dendritic segments were acquired under high magnification (×100/1.40, with immersion oil). The stack of images was then projected along the *z*-axis to obtain a 2D image of the dendrite and the spines. The head contour of each visible spine protruding from the sampled dendrite was traced with FIJI (http://fiji.sc/) and the Feret's diameter (i.e., the longest distance between any two points of the contour) was recorded. Spine head area analysis followed a bimodal distribution (see Fig. 2b) with spines then further categorized as spines with small or large head based on a cutoff value of 0.7 μm², which is the median value of spine head size in WT mice. This criterion for spine classification has been previously validated and it has been shown to reduce the number of false negative in spine analysis[79].

**High-pressure freezing transmission electron microscopy (TEM)**. Animals were perfused with 2.5% glutaraldehyde, 2% formaldehyde, 0.1% Na-cacodylate buffer, (pH = 7.2), after which the brains were removed and kept in the same fixative. Brains were cut in 200-μm-thick slices with vibratome, small discs were punched out of the brain slices and high-pressure frozen in 200 μm-deep carriers in a Leica EMpact2 high-pressure freezer. The samples were freeze-substituted in a Leica AFS2 freeze-substitution apparatus for 24 h at −90 °C in 0.1% tannic acid (Electron Microscopy Sciences), washed 3× in acetone, and further incubated in 2% osmium tetroxide/acetone + 5% H₂O for another 44 h. The temperature was raised during 12 h up to −30 °C. Samples were then washed in acetone 3×, after which the temperature was further raised to 20 °C and the samples were removed from the AFS-apparatus. Samples were detached from the carriers, infiltrated and embedded in epon (Agar100). After polymerization for 48 h at 60 °C, blocks were cut (50–60 nm) with a Leica UCT ultra-microtome, post-stained with 4% uranyl acetate for 10 min and Reynold's lead citrate for 3 min. Finally, sections were imaged with JEOL JEM1400 transmission electron microscope operated at 80 kV and equipped with an Olympus SIS Quemesa (11 Mpxl) camera. Data were analyzed using iTEM software program (Olympus SIS). The PSD length followed a bimodal distribution (see Fig. 2d), therefore PSD was further categorized as short or long using cutoff values of 226.32 and 363.85 nm, which represent the 25 and 75% percentiles of the PSD length measured in WT mice, respectively.

**PSD-AZ visualization by TEM**. Brains were fixed in 100 mM Na-cacodylate, 4% PFA, 2 % glutaraldehyde, and 0.1% picric acid (pH 7.4) and 200-μm-thick slices were sliced with a vibratome. For visualization of the PSD, slices were treated with 1% ethanolic phosphotungstic acid. Subsequently, samples were contrasted in 2% uranyl acetate, washed, dehydrated in a graded series of ethanol solutions, and embedded in epon. One hundred nm sections were counterstained with lead

citrate and examined under the Jeol 1400 electron microscope. The analysis of PSD length and thickness as well as AZ length and synaptic cleft size were performed using ImageJ software by an experimenter blind to the genotype.

**Serial block face scanning electron microscopy (SBF-SEM)**. Animals were perfused with 2.5% glutaraldehyde, 4% formaldehyde, 0.2% picric acid, in 0.1M sodium cacodylate buffer, (pH = 7.4), then brains were removed and kept in the same fixative at 4 °C until further processed. Brains were embedded in 3% agarose and cut in 200-μm-thick slices with a vibratome. The sections were postfixed in 1% osmium tetroxide containing 1.5% potassium ferrocyanide for 30 min at room temperature, stained with 0.2% tannic acid for 20 min, fixed with 1% osmium tetroxide for 30 min, stained with 1% thiocarbohydrazide for 20 min, and incubated again with 1% osmium tetroxide for 30 min. Samples were subsequently contrasted with 0.5% uranyl acetate in 25% methanol overnight at 4 °C and with Walton's lead acetate for 30 min at 60 °C. After ethanol dehydration, the samples were infiltrated and embedded in resin (with the modification of a harder epon replacement mixture; Agar100) as for conventional TEM. A small portion of the barrel cortex was mounted on a pin, pre-trimmed in a microtome. The pin was placed in a scanning electron microscope (Zeiss VP Sigma) equipped with an internal microtome (Gatan, 3View). Serial sectioning was performed at 50-nm steps. Serial backscattered electron images (1.5 or 1.8 kV, 200pA, immersion mode) of the block face, focusing on the region of interest, were recorded at 0.08 μm/pixel resolution. The serial images were aligned using DigitalMicrograph software and reconstructed using Imaris software.

**Subcellular fractionation and immunoblotting**. Briefly, cortices from 7-week-old mice were rapidly dissected and homogenized in 10 volumes of sucrose buffer (0.32 M sucrose, 1 mM HEPES -pH 7.4, 1 mM MgCl₂, 1 mM NaHCO₃, 1 mM EDTA) and protease/phosphatase inhibitor cocktail (PIC) with a glass Teflon Dounce homogenizer, and centrifuged at 800 *g* for 15 min. Homogenates were centrifuged at 900 *g* for 5 min. The resulting supernatant was centrifuged 15 min at 11,000 *g*, and the pellet collected as crude synaptosomal fraction (P2). The P2 fraction was solubilized in 1% Triton buffer (1 mM HEPES, 1% Triton X-100, 75 mM KCl) in the presence of PIC, and centrifuged for 1 h at 100,000 *g*. The pellet (P3) was solubilized in 20 mM HEPES containing PIC, and was considered as triton insoluble fraction (TIF) or PSD-enriched fraction. The supernatant (SN3) was considered as triton soluble fraction (TSF) or presynaptic fraction. Ten μg of protein were subjected to 8–12% sodium docecyl sulfate–polyacrylamide gel electrophoresis (SDS–PAGE) as previously described[80]. The following primary antibodies were used: NR2A (1:2000, Millipore, #07-632), NR2B (1:500, Acris Biosciences, AM20071PU; 1:1000, Millipore, #06-600), GluR1 (1:1000, Millipore, MAB2263), GluR2 (1:2000, Chemicon, AB1768), mGluR5 (1:1000, Millipore, AB5675), PSD-95 (1:3000, Thermo Scientific, MA1-045), synaptophysin (1:500, Abcam, ab8049), GABA_A α1 subunit (1:500, Upstate, #06-868), Homer (1:200, Santa Cruz, sc17842), Arc (1:500, Synaptic systems, #156002), phospho-αCaMKII (Thr286) (1:1000, Cell Signaling, #12716), αCaMKII (1:1000, Millipore, #05-532), FMRP-rAM2 (1:1000, produced in our laboratory[10]), eIF4E (1:1000, Cell signaling, #9742), Gephyrin (1:1000, Synaptic systems, #147111), ubiquitin (1:200, Dako, Z0458) and actin (1:5000, Sigma, A5441). The membranes were scanned using an Odyssey infrared imager (LI-COR Biosciences), and protein levels were quantified using Image Studio Lite Version 5.2 (LI-COR Biosciences). Images have been cropped for presentation, full size images used in the different Figures can be found in Supplementary Fig. 6.

**Preparation of cortical synaptoneurosomes**. Synaptoneurosomes were prepared from 7–8-week-old WT and *BC1* KO mice. Cortices were rapidly dissected and placed into homogenization buffer (0.32 M sucrose, 5 mM HEPES -pH 7.4-, 2 mg/ml BSA, 1 mM EDTA). Cortices were homogenized in a 7 ml Kontes tissue Dounce homogenizer with 10 strokes. Homogenized tissue was centrifuged at 3000 *g* for 8 min at 4 °C, and supernatant centrifuged at 14,000 *g* for 12 min at 4 °C. Crude synaptosomal fraction was resuspended in Krebs ringer buffer (140 mM NaCl, 5 KCl, 5 mM D-glucose, 10 mM HEPES –pH 7.4-, 1 mM EDTA), and mixed gently with equal volumes of Percoll™ (GE Healthcare). After centrifugation, synaptoneurosomes were recovered, washed twice with Krebs ringer buffer and resuspended in Hepes-Krebs buffer (147 mM NaCl, 3 mM KCl, 2 mM MgSO₄, 3 mM CaCl₂, 10 mM D-glucose, 20 mM HEPES -pH 7.4-). Treatments with cycloheximide (25 μM) and 4EGI (50 μM) were for 1 h at 37 °C with gentle rocking, and then synaptoneurosomes were washed twice with Hepes-Krebs buffer and resuspended in lysis buffer (Tris-HCl -pH 7.4-, 150 mM NaCl, 1% triton X-100) containing PIC. Twenty μg of protein from each sample was subjected to 8–12% SDS–PAGE and western blotting as described above.

For determination of ubiquitinated proteins, synaptoneurosomes were prepared using the same procedure with the addition of MG132 (10 μM) to homogenization and lysis buffers.

**Metabolic labeling of synaptoneurosomes**. Detection of de novo protein synthesis was determined by using metabolic labeling with Click-iT L-azidohomoalanine (AHA, Molecular Probes), as previously described[36] with minor modifications. Cortical synaptoneurosomes were pre-incubated with 500 μM

AHA in Hepes-Krebs buffer for 50 min at 37 °C, and then treated with 100 µM DHPG for 10 min. After washing out the treatments, samples were solubilized in lysis buffer containing PIC and incubated with 500 µM phosphine-PEG3-Biotin (Thermo) for 4 h at 37 °C to conjugate biotin to AHA-containing proteins. Samples were then eluted on Zeba Spin desalting columns (7K MWCO, Thermo) to remove the excess of phosphine-PEG3-Biotin. Equal amount of proteins (~ 5 mg/ml) were incubated with PSD-95 antibody (2 µg, Thermo) overnight at 4 °C. Subsequently, 30 µl of protein G-dynabeads (Thermo) were added to each sample and incubated for 45 min at 4 °C with gentle rocking. After three washes, samples were processed for SDS–PAGE and western blotting. IRDye 800CW Streptavidin (1:2000, LI-COR Biosciences) was used to detect biotin-conjugated (newly synthesized) PSD-95. Anti- PSD-95 antibody (1:3000, Thermo, MA1-045) was used to detect total immunoprecipitated PSD-95.

**F/G-actin assay**. The relative amount of F-actin and G-actin was determined using the F/G-Actin In Vivo Assay kit (Cytoskeleton Inc., BK037) according to manufacturer instructions. Briefly, 30–50 mg of cortical tissue was homogenized in Lysis and F-actin Stabilization Buffer (cat # LAS01) containing 1 mM ATP and PIC. Homogenates were incubated for 10 min at 37 °C, and centrifuged at 350 $g$ for 5 min. Supernatants were then ultracentrifuged at 49,000 $g$ for 1 h at 37 °C. The resulting supernatant was collected as G-actin fraction, whereas the pellet corresponding to the F-actin fraction was resuspended in F-actin depolymerizing buffer (cat # FAD02). Samples were then subjected to 12% SDS–PAGE and analyzed by immunoblot using anti-actin antibody (1:500, Cytoskeleton).

**Fluorescent in situ hybridization (FISH)**. After decapitation, the cortex from WT mice was dissected, snap-frozen in isopentane and then stored at −70 °C until being sectioned on a cryostat. Twenty micrometer-thick transversal or coronal sections were mounted on slides. Slices were treated with 4% Parafomaldeyde for 30' and washed with saline-sodium citrate (SSC) buffer 0,5X twice for 10 min. The slices were then treated with Proteinase K solution (7 µg/µL Proteinase K, 0,5 M NaCl, 10 M Tris-HCl, pH 8.0) for 30 min and then treated with pre-Hybridization solution (50% formammide, 4X SSC, 1X Denhardt's Solution, 2 mM EDTA, 500 µg/ml ssDNA) for 2 h at 42 °C. Digoxigenin-labeled $BC1$ RNA antisense was hybridized on tissue sections overnight at 53 °C and detected with the use of anti-digoxigenin–HRP conjugate (Roche) and a commercial cyanine-5 tyramide signal amplification kit (TSA-CY3; PerkinElmer). Sections were counterstained for nuclei with Hoechst (Hoechst; Life technologies). The specificity of the labeling was monitored by omitting the riboprobe or using $BC1$ sense RNA probe. Images were acquired using an inverted confocal microscope (Zeiss).

$BC1$ RNA sense and antisense digoxigenin-labeled riboprobes sequences are:
AS Probe Exiqon: Probe: /5DigN/AAAGGTTGTGTGTGCCAGTTA/3DigN/
Position in target ($BC1$ sense): 134–154
Sense Probe Exiqon: Probe: /5DigN/AAACAAGGTAACTGGCACACA/3DigN/
Position in target ($BC1$ sense): 126- 146

**RNA purification and qPCR**. Total RNA from somatosensory cortex was extracted using TRIzol (Invitrogen). Eluted RNA was reverse transcribed, and qPCR was performed using LightCycler 480 SYBR Green I (Roche Applied Science). Primer sequences:
$BC1$ forward: 5′-CTGGGTTCGGTCCTCAG-3′
$BC1$ reverse: 5′-TGTGTGTGCCAGTTACCT-3′
PSD95 forward: 5′-CATTGCCCTGAAGAACGC-3′
PSD95 reverse: 5′-ATGGATCTTGGCCTCGAA-3′
CamK2a forward: 5′-GTGCTGGCTGGTCAGGAGTATGC-3′
CamK2a reverse: 5′-CTTCAACAAGCGGCAGATGCGGG-3′
Hprt1 forward: 5′-CAGCCCCAAAATGGTTAAGGTTGC-3′
Hprt1 reverse: 5′-TCCAACAAAGTCTGGCCTGTATCCA-3′
Gusb forward: 5′-CCGCCGCATATTACTTTAAGAC-3′
Gusb reverse: 5′-CCCCAGGTCTGCATCATATTT-3′

**Primary neuronal cultures and transfection**. Mouse primary cortical neurons were prepared at embryonic day 15 (E15) as previously described[53]. For spine morphology analysis, neurons at day in vitro (DIV) 7 were transfected with EGFP-containing construct using lipofectamine 2000 (Invitrogen). At DIV14, neurons were fixed in PFA solution (4% PFA, 0.12 M sucrose, 3 mM EGTA, 2 mM MgCl$_2$ in phosphate-buffered saline (PBS)).

**Dendritic branching and spine morphology analysis in vitro**. For analysis of dendritic arbors, images of GFP-transfected neurons were taken using an Olympus IX71 widefield fluorescence microscope (×20 objective) at 2048 × 2048 pixels resolution. Neuronal arbors were traced and subjected to Sholl analysis, using NeuroJ and Sholl analysis plugins, respectively, within ImageJ software (NIH, Bethesda, MD, USA).

For spine analysis, confocal images were obtained using a confocal laser scanning microscope (Nikon A1R Eclipse), ×60 oil objective, with a sequential acquisition setting at 2048 × 2048 pixels resolution. Spine analysis was performed in primary and secondary dendrites following previously described protocols[51].

Quantitative analysis was performed by two independent experimenters, each one analyzing different sets of data.

**Biotinylation assay**. Cortical neurons (DIV 14) were washed in cold PBS and incubated with or W/O 0.2 mg/ml Sulfo-NHS-LC-Biotin (Thermo Scientific) in PBS for 30 min at 4 °C. Excess biotin was quenched with 200 mM glycine. Cells where then lysed in STEN buffer (150 mM NaCl, 50 mM Tris-HCl pH 7.4, 2 mM EDTA, 1% NP-40, 1% Triton X-100) containing protease inhibitor cocktail (Sigma), and the biotinylated proteins pulled down with streptavidin Dynabeads (Invitrogen) for 1 h at 4 °C. Proteins were then separated by SDS–PAGE analyzed by Western Blotting.

**SUnSET assay**. Basal protein translation was measured by using the SUnSET assay as previously described[35]. Briefly, WT and $BC1$ KO cortical neurons (DIV14) were treated with puromycin (10 µg/ml) for 45 min, and then lysed in lysis buffer (150 mM NaCl, 50 mM Tris-HCl pH 7.4, 1% Triton X-100) containing PIC. Neurons treated with cycloheximide (50 µM) 15 min before puromycin and neurons not treated with puromycin were used as negative controls. Twenty µg of protein were subjected to SDS–PAGE and analyzed by western blotting using an anti-puromycin antibody (1:500, DHSB, PMY-2A4). Ponceau red staining was used as protein loading control.

**Whisker deprivation and immunoblotting**. Six-week-old mice were subjected to unilateral whisker trimming or left untreated (control). Seven days later brains were collected for morphological (Golgi staining) or biochemical (western blot) analysis.

Somatosensory cortex from each hemisphere (ipsilateral and contralateral) of WT and $BC1$ KO mice was micro-dissected and homogenized in lysis buffer (100 mM NaCl, 10 mM MgCl$_2$, 10 mM Tris-HCl [pH 7.5], 1 mM dithiothreitol, 30 U/ml RNasin, 1% Triton X-100, 0.5 mM Na-orthovanadate, 10 mM β-glycerophosphate, and 10 mg/ml Sigma protease inhibitor cocktail), incubated 5 min on ice, and centrifuged at 12,000 $g$ for 8 min at 4 °C. Ten µg of the supernatants were loaded into 4–12% SDS–PAGE. The following antibodies were used: vinculin (1:1000; Sigma; V9131), NR2A (1:1000, Millipore, #07-632), anti mGluR5 (1:1000, Millipore, AB5675), GluR1 (1:1000, Millipore, MAB2263). The secondary horseradish peroxidase-conjugated antibodies (Amersham) were used 1:5000. The signal, detected using an ECL plus or advanced detection system (Amersham), was captured by a Fujifilm LAS-3000 digital camera and quantified by means of Aida software (Raytest Isotopenmeßgeräte GmbH).

**Behavior**. Six-to-15-week-old $BC1$ WT and KO (C57Bl/6 background) male mice were used for behavior. Animals were housed in groups of 2–4 per cage in a flow-cabinet with controlled temperature and humidity. Food and water was provided ad libitum and cotton cylinders or tissue were used as cage basic enrichment for nest building. The dark-light cycle was normal (with lights on from 7.00 to 19.00 hours and off from 19.00 to 7.00 hours). Before the start of the behavioral experiments, all mice were allowed to habituate for 1 week to the flow-cabinet. Testing took place in a room next to the housing room outside the flow-cabinet.

**Novel object recognition test (NORT)**. An extended NORT consisting of two test phases (i.e., visual and texture NORT) was performed according to a previously published protocol[46]. During the first 2 days of behavioral testing, mice were subjected to 10-min sessions in an open field arena with opaque walls (45 × 45 × 45 cm) to habituate the animals to the environment before the tests. The bottom of the arena was covered with bedding material. Mice activity was tracked with Ethovision software (Noldus, Wageningen, The Netherlands). Distance traveled and time spent in the center area of the arena was measured. Between each mouse, the bedding material was thoroughly mixed to avoid scent trails guiding subsequent animals.

On the third day, a standard NORT (i.e., visual NORT) was implemented which consists of two phases. In the first phase, the mouse was released at the edge of the arena, and equidistant from two identical objects (Familiar; two 50 ml Falcon tubes containing water). The objects were placed equidistant from the center of the arena, and equidistant from the arena walls. Each mouse was allowed to freely explore the objects for 10 min. Animals were afterwards removed from the test arena for 5 min. During this retention period, one object was replaced with a new object, which was visually distinct from the other object (Novel; 50 ml Falcon tube wrapped in red paper). In the second phase, the mouse was placed back into the test arena and allowed to freely explore the objects for 10 min. The time that each mouse spent physically exploring the objects was measured during both phases. The location of the novel object was counter-balanced among animals.

On the fourth day, a texture-based NORT was implemented. During the first phase, the mouse was allowed to freely explore two objects that were identical in texture (Familiar; two 50 ml Falcon tubes wrapped with sandpaper gradient 80) for 10 min. After 5 min retention period, the mouse was reintroduced in the test arena. During the second phase, one of the objects was replaced by a visually identical but texturally different object (Novel; 50 ml Falcon tube wrapped with sandpaper gradient 100) and mouse exploration behavior towards the two objects was

measured. The location of the novel object and the texture of the familiar object was counter-balanced among animals. The novelty preference index was calculated as: $\text{Time}_{novel} / (\text{Time}_{familiar} + \text{Time}_{novel})$. Animals that did not explore the two objects for at least 5 s were not included in the analyses.

**Sociability/social novelty**. Sociability and social memory were evaluated using the three-chamber test. The set-up consists of a rectangular transparent plexiglass box divided into three compartments, separated by two partitions. These partitions contain a small passage in the middle to allow the mouse to move between compartments and can be closed by small sliding doors. During the habituation phase, mice are allowed to explore the empty central compartment for 5 min. At the start of the test, both sliding doors are closed and each of the outer compartments are equipped with a wired cage (about 10 cm Ø, 15 cm high), one empty and the other containing a mouse, now referred to as 'stranger1'. These cages are used to avoid inter-male fighting but allow visual and olfactory contact between mice. The tested mouse is placed in the closed middle compartment and the sliding doors are removed to allow the mouse to move through the test freely. After 10 min of free exploration, the mouse is placed back into an empty cage for 5 min. 'Stranger 1' mouse becomes the familiar mouse during the social novelty paradigm. Another unfamiliar mouse is placed under the second wired cage, now referred to as 'stranger 2'. Again, the sliding doors are removed and the mouse is allowed to explore for 10 min. Location of 'stranger 1' and 'stranger 2' mouse was counter-balanced between animals and the box was cleaned with soapy water after each animal. All stranger mice used were of the same sex as the tested mouse and had never been housed in the same animal room. Mice were recorded with a camera and their heads tracked using ANY-Maze software (Stoelting Europe). Time spent in close proximity ( < 2 cm) to the cages was monitored and manually scored by an experimenter blind to the genotype. Preference index was calculated as: $\text{Time}_{stranger1}/(\text{Time}_{stranger1} + \text{Time}_{empty})$ and $\text{Time}_{stranger2}/(\text{Time}_{stranger1} + \text{Time}_{stranger2})$ for sociability and social memory, respectively.

**Self-grooming**. Animals were individually placed in a clear observation cylinder (13 cm in diameter, 20 cm height), and self-grooming activity monitored using one top and one front cameras for 5 min. Time spent grooming was manually scored by two independent experimenters blind to the genotype. Each experimenter analyzed videos recorded with different cameras, and data obtained from both cameras was averaged for each animal.

**Marble burying test**. The set-up consisted of a large cage ($45 \times 22$ cm) with filter tops filled with 5 cm of fresh and compact bedding and containing 24 marbles ($\sim 1.5$ cm diameter each) equally distributed along the walls (at 2 cm distance). Mice were placed in the cage and left undisturbed for 30 min. At the end of the test, the number of marbles buried more than 2/3 of their volume were counted by an experimenter blind to the genotype.

**Nest building test**. Animals were housed individually and provided with three cotton cylinders (0.6 g each) and left undisturbed for 24 h. Unused cotton cylinders were weighed and nest quality was scored by an experimenter blind to the genotype according to a previously described protocol[51].

**Automated tube test**. The automated tube test was performed as described previously[81], with minor modifications. Mice (5–6 week old) were housed in WT/KO pairs. Experiments started few weeks later (i.e., when mice were 12–15 week old). Matches were performed between all mice regardless of the genotype and home-cage origin. A match was considered completed when both mice entered the goal box relative to the winner (or starting box relative to the loser) with all four paws.

**Statistical analyses**. Comparisons between the two groups were performed using unpaired Student's $t$-test or non-parametric Mann–Whitney $U$-test. One-way or two-way analysis of variance followed by a post-hoc Newman-Keuls or Bonferroni's multiple comparison tests were used when more than two groups were compared. Distributions were analyzed using the Pearson's chi-square ($\chi^2$) test. For molecular/electrophysiological studies, sample size was not estimated prior to the experiments, we believe it is adequate to measure the effect size based on the literature. For behavioral studies, we estimated requiring a total sample size of 9 mice per genotype based on power analysis (power equal to 80%, alpha equal to 0.05 and effect size equal to 1.5 x SD, based on previous behavioral studies from our laboratory). Results were presented as mean ± standard error of the mean (s.e.m.) from the indicated number of independent experiments and expressed as fold of the indicated control. $P$ values > 0.05 were regarded as not significant. Data points lower or higher than 1.5 x IQR (interquartile range) were considered outliers and kept in the figure but excluded from the statistical analysis. Samples were generally randomized according to the tag number code of the animals or a simple number code for in vitro experiments.

**Data availability**. All relevant data are available from the authors upon request.

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

## Acknowledgements

We are thankful to Eef Lemmens and Annick Crevoisier for excellent administrative support and to Jonathan Royaert, Karin Jonkers, Joanna Viguie and Nathalie Leysen for technical assistance with neuronal cultures and mouse colonies. We thank Frone Vandewiele for helping with the membrane fractionation experiments and Ria Van Laer for technical assistance with the CO histochemistry. We are grateful to Zsuzsanna Callaerts-Vegh, Hilde Brems and Rudi D'Hooge for valuable discussions of the behavioral experiments. Finally, we thank Eleonora Rosina, Marijke Laarakker, Martien Kas, Muna Hilal, Carl Petersen, Sermet Berat Semihcan for the preliminary electrophysiological and behavioral data and Egbert Welker and Tilmann Achsel for critical reading of the manuscript, constructive experimental suggestions and data analysis. The Nikon microscope used in this study was acquired through a Hercules Type 1 AKUL/09/037 to Wim Annaert. This work was supported by the Fonds Wetenschappelijk Onderzoek Vlaanderen (FWO G070511N10 and GO88415N), Telethon (GGP15257), Queen Elisabeth Foundation (FMRE), Solvay Price, VIB, NEURON ERA-NET and 51NF40-158776 NCCR Synapsy to C.B.; Dutch Medical Research Council, NWO ZonMW (# 91710372) to R.M.; EU-BrainTrain (FP7-People-ITN-2008-238055) to C.B. and R.M.; German Research Foundation (SFB 874/A9) to P.K.

## Author contributions

V.B., E.P. and L.R. performed the spine analysis in vitro and in vivo. K.J. performed the in vivo recordings of barrel cortex activity. The A.B. and T.G. performed the in situ hybridization experiments. R.L. performed the biotinylation experiment. R.P. and R.M. performed and analyzed the spontaneous activity data in vivo. N.V.G., V.B. and E.P. performed and analyzed the EM data. P.B. provided technical assistance with the EM experiments. V.B. analyzed the expression level of receptors, postsynaptic proteins, performed, the Click-it experiment and the nucleotide alignments. V.B. and E.P. performed and analyzed the puromycin experiments. L.R. and V.M. performed the whiskers deprivation studies and analyzed the receptor levels. J.N., E.P., R.L. and L.A. performed and analyzed the CO staining. A.C.L. and V.B. performed all the behavioral tests, scored the self-grooming experiment independently and analyzed all the behavioral experiments independently. H.M., G.F., M.A.-T., P.K. and C.B. supervised the different subprojects. C.B. directed the research. V.B. and C.B. wrote the manuscript and all the authors contributed to the writing.

## Additional information

**Competing interests:** The authors declare no competing financial interests.

