## [Peer Review File · Nature Communications]

Reviewers' comments:

Reviewer #1 (Remarks to the Author):

This manuscript describes the consequences of BC1 knockout on dendritic complexity, dendritic spine morphology, post-synaptic density size and makeup, neuronal activity, and behavioral phenotypes in mice. The authors were able to show that knockout of BC1 in mice, while decreasing dendritic complexity, increases spine density in layer 2/3 neurons and layer 4 stellate cells but not in layer 5 neurons. They were also able to show an effect on PSD length and thickness. These findings align well with the results from electrophysiological experiments, which show a higher spontaneous spiking activity, as well as with the comparisons in quantities of post-synaptic proteins involved in neuronal excitation and plasticity. Abnormal plasticity in the barrel cortex was also confirmed via whisker deprivation. Finally, they showed differential novel object preference, sociability, and grooming behavior between WT and BC1 KO mice. This is a well-written manuscript with significant implications concerning the relationship between neuronal plasticity and social behavior, and describes possible mechanisms that underlie these processes via the ncRNA BC1.

Major weakness:

1. The connection between latency to investigate and barrel cortex plasticity in particular is not clear since the mice do not get sensory information about the object via the whiskers until they are in contact with the object. An analysis of the time spent with each object might be more helpful in determining object preference. Furthermore, suggesting deficits in object discrimination is highly speculative since memory of the object and/or preference for novel objects may be affected.

Minor weakness:

1. Authors need to provide more information about how the BC1 KO mice was generated, the breeding strategy, and which experiments were conducted with mixed background.
2. It is very difficult to visualize the expression of BC1 RNA in nucleus and cytoplasm in supplementary figure 1, please provide better images.
3. An interpretation of the increased variability in spiking frequency, average response to PW, and average response to SW in BC1 KO mice would help convey the importance of these results.
4. In line 425-426 "In contrast, whisker trimming attenuates spine loss in layer 5 pyramidal neurons" it is unclear what the attenuation is in response to. Deletion of Fmr1?
5. In the SUnSET assay, it is difficult to visualize the coomassie blue staining. Please provide clearer images or normalize to a housekeeping gene.
6. In line 255 similar is written twice.
7. The background information on prior studies of BC1 KO mice is relatively superficial, given the substantial amount of work that has been done using this model.

Reviewer #2 (Remarks to the Author):

In this study, Briz and the colleagues have performed the analysis of BC1 knock-out mice to determine the role of BC1 in the barrel cortex. Because BC1 forms ribonucleoprotein particles with different protein partners including the Fragile X Mental Retardation Protein, the elucidation of BC1 functions, especially focusing on the relevance with neuropsychiatric phenotypes, is of great importance. Indeed, I think this is a very important piece of work. However, I do have some major concerns in regards of the experimental design and data interpretation.

Major points

(1) Methods section describes that "In some experiments, 6-8 week-old WT and BC1 KO mice (129SV-C57/Bl6 mixed background) were also used". If 129 mice are included in the behavioral analysis, please exclude all data of 129 mice from behavioral analysis.

(2) Findings from in vivo electrophysiology experiment (layer 2/3 pyramidal neurons) is quite interesting given that the alternation of the dendritic spine properties (size and density) in KO mice. Although the authors performed the layer specific analysis of the spine density, spine size analysis seems to be pooled data of layer 2/3 and 5 pyramidal neurons. To understand the in vivo e-physics data (layer 2/3 pyramidal neurons) properly, layer 2/3 pyramidal neuron-specific analysis of the spine size must be performed. Also, related to the in vivo e-physics, it is well known that the anesthesia has a great impact on the firing pattern. It should be guaranteed that the depth of the anesthesia is indeed identical between genotypes. The detailed and quantitative monitoring to assess the depth of anesthesia is necessary.

(3) Figure 4a: the author clearly demonstrated the increased protein level in the NR2B/GluR1/GluR2/PSD-95, which is consistent with structural and functional property of excitatory synapse. Although these data are convincing, it would be useful information whether such effect is limited in the excitatory synapse or is also observed in the inhibitory synapse. The authors demonstrated that the expression level of GABA_A1 was not affected. However, the detection of GABA_A1 alone is not sufficient to conclude inhibitory synapse is not affected. Perhaps, it is better to demonstrate the expression of Gephyrin.

(2) In line 242, the authors described that "In brief, a modified aminoacid, L-azidoalanine (L-AHA), was incorporated into nascent proteins and subsequently biotinylated using the Click-It technology²⁷. An overall increase in L-AHA incorporation was found in BC1 KO mice as compared to WT mice (Fig. 4d)". Because the discrete gels were used for the biotin-positive immunoblot for WT and KO, such comparison between WT and KO is not appropriate. The authors need to perform the immunoblot of WT and KO on the same gel.

(3) In the figure 4e, the authors claimed that a de novo-synthesised PSD-95 increased upon DHPG treatment in WT, but not in KO mice. However, it is quite difficult to see the band at 95 kDa for streptavidin blot (de novo-synthesised PSD-95). In addition, bigger and smaller bands than 95 kDa were also observed. What are those bands? The blotting needs to be replaced for better presentation.

(4) Figure 5c: A significant decrease in NR2A, but not GluR1 or mGluR5, was detected in the contralateral cortex of WT mice. Given that the spine density change was obvious in the contralateral cortex of WT mice (Fig 5b), NR1A specific decrease seem to be odd. This point should be carefully discussed.

(5) Figure 6a-b: NORT is often used to detect the recognition memory as well as novelty seeking behaviors. In this regard, the longer latency to first contact with the novel object compared to the familiar one is not necessarily indicative of the inability to discriminate between the new and familiar, but it can reflect the decrease in the novelty-seeking traits. Thus, the statement (line 307) that "indicating the inability of BC1 KO mice to discriminate between the two objects" is overstatement. If the authors would like to claim the recognition memory deficit, it should be demonstrated that novelty-seeking trait is intact in KO mice. Related to above, my confusion stems from the sentence in line 316 that "These results support the idea that defects in novel object recognition are not due to deficits in their ability to discriminate novelty, but they are rather specific to the task, possibly involving impaired processing of sensory information". What does this sentence mean? The manuscript needs to be revised in a logical manner.

(6) Line 337: the authors stated that "our results indicate that BC1 KO mice have deficits in object discrimination, altered social behavior and engage more frequently in self grooming, behavioral phenotypes commonly observed in animal models of ASD, including FXS". Typical ASD model mice also exhibit the altered phenotype of social interaction test (Fig 6d), Marbles test (Fig 6H), and nest-building behaviors (Fig i, j). Since the authors' data are important findings, these must be discussed carefully: how are BC KO mice different from other ASD model mice, especially FMR KO and CYFIP1 KO mice.

Minor points

(1) The information about the antibodies is insufficient. Please provide enough information so that readers can identify the antibodies that were used.

(2) Please provide much detailed information of spine head size analysis.

(3) In some experiments, it is unclear how many mice/neurons/spines were analyzed in each experiment; N values are not sufficiently given. Can the authors please provide these information in the figure legends (or in the new supplemental figure) along with their statistic values?

Reviewer #3 (Remarks to the Author):

In this manuscript, the authors investigated roles of the brain cytoplasmic (BC1) RNA on cortical functions. Using BC1 knock out mice, they showed the overall increase of spine density, increased spine size, and larger excitatory postsynaptic currents. They also reported altered structural plasticity of the barrel cortex and impaired behaviors of BC1

knock out mice. BC1 has been known to function with the Fragile X Mental Retardation Protein, FMRP, and suggested to be related to the Autistic Spectrum Disorders (ASD). Thus, the characterization of BC1 in the cortical function is a crucial issue in the field. Although this manuscript investigated this important issue, the current manuscript is still immature and requires further experiments and discussion.

Major

- 1) The authors showed that while the cortical spine density is increased in the BC1 knock out mice, the frequency of sEPSC is unchanged. The authors should explain this issue.
- 2) The circuit and information flow in the barrel cortex has been well characterized in the previous reports. The authors should examine evoked mEPSC to further elucidate the cortical alteration of BC1 knock out mice.
- 3) In the Figure 3c, the authors showed no change in the frequency of sEPSC of mutant barrel cortex. However, in vivo recording exhibited increased spontaneous spikes. The authors should explain this issue.
- 4) Using in vivo recording, the authors should further investigate the functional whisker-barrel map (Stern et al. 2001, Jitsuki et al. 2011). This is important for relating synaptic function, circuit, and behaviors.
- 5) In the Figure 4, the authors showed the alteration of NR2B, GluR1, GluR2, and mGluR5 in the mutant barrel cortex. On the other hand, only NR2A is changed in the whisker deprivation condition in the mutant barrel cortex compared to wild type.
- 6) What is a mechanisms underlying the regulation of the expression of NR2A with BC1?
- 7) They showed the abnormality of object recognition and social recognition. Since these behaviors could also be regulated by other brain regions than the barrel cortex, the authors should thoroughly test other learning tasks.

Minor

- 1) In the Figures 4a,b, they used actin for the normalization. Why did they use vinculin for the normalization in Figure 5?
- 2) There is a TypeO at line 255.

Answers to reviewers' comments

Reviewer #1 (Remarks to the Author):

This manuscript describes the consequences of BC1 knockout on dendritic complexity, dendritic spine morphology, post-synaptic density size and makeup, neuronal activity, and behavioral phenotypes in mice. The authors were able to show that knockout of BC1 in mice, while decreasing dendritic complexity, increases spine density in layer 2/3 neurons and layer 4 stellate cells but not in layer 5 neurons. They were also able to show an effect on PSD length and thickness. These findings align well with the results from electrophysiological experiments, which show a higher spontaneous spiking activity, as well as with the comparisons in quantities of post-synaptic proteins involved in neuronal excitation and plasticity. Abnormal plasticity in the barrel cortex was also confirmed via whisker deprivation. Finally, they showed differential novel object preference, sociability, and grooming behavior between WT and BC1 KO mice. This is a **well-written manuscript with significant implications concerning the relationship between neuronal plasticity and social behavior**, and describes possible mechanisms that underlie these processes via the ncRNA BC1.

We thank the Referee for the positive comments on our manuscript.

Major weakness:

1. The connection between latency to investigate and barrel cortex plasticity in particular is not clear since the mice do not get sensory information about the object via the whiskers until they are in contact with the object. An analysis of the time spent with each object might be more helpful in determining object preference. Furthermore, suggesting deficits in object discrimination is highly speculative since memory of the object and/or preference for novel objects may be affected.

We agree with the Reviewer that the time spent with each object is more relevant than the latency to first contact with the object for determining object preference. Following his/her advice we have now analyzed the time spent with each object, and results are now shown in the *revised Suppl. Figure 5*. We have also changed object discrimination for object preference in the text.

Minor weakness:

1. Authors need to provide more information about how the BC1 KO mice was generated, the breeding strategy, and which experiments were conducted with mixed background.

We have added the requested information in the methods section. All experiments were performed in KO mice of two different genetic background (C57/Bl6 and 129SV-C57/Bl6), except for the behavioral experiments, which were done in C57/Bl6 mice only. The mice were generated in the laboratory of prof. Jurgen Brosius and were already described in several publications from his group as well as in Napoli et al., 2008, Lacoux et al 2012.

2. It is very difficult to visualize the expression of BC1 RNA in nucleus and cytoplasm in supplementary figure 1, please provide better images.

Following the Reviewer's request we have performed a new data set of FISH experiments that we have added in the revised Suppl. Figure 1; the subcellular expression of BC1 RNA in cells from the barrel cortex is now clearer.

3. An interpretation of the increased variability in spiking frequency, average response to PW, and average response to SW in BC1 KO mice would help convey the importance of these results.

We thank the Reviewer for the suggestion. Neuronal encoding of sensory information relies in part on comparing spiking between PW and SW responses (Higley and Contreras, 2007; Adibi et al 2013; Scaglione et al 2014). The increased spiking variability and the ensuing changes in temporal information seen in the BC1 KO mice could thus contribute to deficiencies in a tactile dependent, novel object recognition task, where the animals use their whiskers to explore objects with different shapes and textures. These comments have been added to the discussion (pg 15).

4. In line 425-426 "In contrast, whisker trimming attenuates spine loss in layer 5 pyramidal neurons" it is unclear what the attenuation is in response to. Deletion of Fmr1?

We apologize if the sentence was not clearly written. The attenuation is in response to whisker trimming, as compared to non-deprived WT mice. We have revised the sentence to clarify this point (pg 15).

5. In the SUnSET assay, it is difficult to visualize the coomassie blue staining. Please provide clearer images or normalize to a housekeeping gene.

Following the Referee's request we have provided a new dataset with different images for the SUnSET assay. In this case Ponceau staining has been used as normalizer in this experiment. Data are now shown in new Figure 5.

6. In line 255 similar is written twice.

Thank you! This has been corrected.

7. The background information on prior studies of BC1 KO mice is relatively superficial, given the substantial amount of work that has been done using this model.

As briefly mentioned above, the mice were generated in the laboratory of prof. Jurgen Brosius and already described in several publications from his group as well as from our group (Napoli et al., 2008, Lacoux et al 2012). We have now included more references and further description of the BC1 KO mouse in the introduction and methods sections. We hope to have now covered all the relevant publications, not many, regarding the role of BC1 RNA in synaptic function and behavior.

Reviewer #2 (Remarks to the Author):

In this study, Briz and the colleagues have performed the analysis of BC1 knock-out mice to determine the role of BC1 in the barrel cortex. Because BC1 forms ribonucleoprotein particles with different protein partners including the Fragile X Mental Retardation Protein, **the elucidation of BC1 functions**, especially focusing on the relevance with neuropsychiatric phenotypes, **is of great importance**. Indeed, **I think this is a very important piece of work**. However, I do have some major concerns in regards of the experimental design and data interpretation.

We thank the Reviewer for the positive comments on our manuscript.

Major points

(1) Methods section describes that "In some experiments, 6-8 week-old WT and BC1 KO mice (129SV-C57/Bl6 mixed background) were also used". If 129 mice are included in the behavioral analysis, please exclude all data of 129 mice from behavioral analysis.

Following reviewer's suggestion we have re-analyzed the data generated from the two different strains separately. Interestingly, while both BC1 KO strains have impaired novel object recognition and reduced social behavior, only 129SV-C57/Bl6 (but not C57/Bl6) show increased self-grooming and decrease marble burying, possibly indicating increased anxiety in these mice, a result consistent with a previous study-ref. 6- in which 129SV-C57/Bl6 mice were used. Due to these differences and following reviewer's suggestion, we are now showing in *revised Figure 7* behavioral data from C57Bl6 mice only and briefly discussed the genotype-difference in the text.

(2) Findings from in vivo electrophysiology experiment (layer 2/3 pyramidal neurons) is quite interesting given that the alternation of the dendritic spine properties (size and density) in KO mice. Although the authors performed the layer specific analysis of the spine density, spine size analysis seems to be pooled data of layer 2/3 and 5 pyramidal neurons. To understand the in vivo e-physics data (layer 2/3 pyramidal neurons) properly, layer2/3 pyramidal neuron-specific analysis of the spine size must be performed. Also, related to the in vivo e-physics, it is well known that the anesthesia has a great impact on the firing pattern. It should be guaranteed that the depth of the anesthesia is indeed identical between genotypes. The detailed and quantitative monitoring to assess the depth of anesthesia is necessary.

We are sorry that our initial explanation was not clear. Spine head size was indeed analyzed on layer 2/3 pyramidal neurons only (pg 37, figure legend).

Concerning the depth of anesthesia, we agree with the Reviewer about such a concern and we have now provided more information about how it was monitored in the methods section. Specifically we used a carefully titrated dosage of urethane based on animal weight. In addition breathing and spontaneous whisker movements were monitored and kept constant. We have also regularly checked hind leg withdrawal and corneal (blinking) reflexes. Therefore, based on visible physiological signs the animals were in the same depth of anesthesia. Furthermore, we chose urethane because it is known for

producing a long-lasting steady level of anesthesia (Maggi & Meli (1986), doi:10.1007/BF01952426.; Hara & Harris (2002), doi: 10.1213/00000539-200202000-00015) that was necessary for our type of experiments.

(3) Figure 4a: the author clearly demonstrated the increased protein level in the NR2B/GluR1/GluR2/PSD-95, which is consistent with structural and functional property of excitatory synapse. Although these data are convincing, it would be useful information whether such effect is limited in the excitatory synapse or is also observed in the inhibitory synapse. The authors demonstrated that the expression level of GABA_A1 was not affected. However, the detection of GABA_A1 alone is not sufficient to conclude inhibitory synapse is not affected. Perhaps, it is better to demonstrate the expression of Gephyrin.

Following the Reviewer's advice we have analyzed the expression of Gephyrin in PSD-enriched preparations, and results have been included in *revised Figure 4*. No changes were observed in this case.

(2) In line 242, the authors described that "In brief, a modified amino acid, L-azidoalanine (L-AHA), was incorporated into nascent proteins and subsequently biotinylated using the Click-It technology²⁷. An overall increase in L-AHA incorporation was found in BC1 KO mice as compared to WT mice (Fig. 4d)". Because the discrete gels were used for the biotin-positive immunoblot for WT and KO, such comparison between WT and KO is not appropriate. The authors need to perform the immunoblot of WT and KO on the same gel.

This is absolutely an important point. The immunoblot performed to analyze nascent proteins from WT and KO was performed on the same gel, but with additional samples in between and therefore only some of the lanes were selected for the final image. We are now providing for the Reviewer only, an image of the entire and original gel. Lanes 1-3 and 8-10 were used to assemble Figure 5. We have added this information in the figure legend.

(3) In the figure 4e, the authors claimed that a de novo-synthesised PSD-95 increased upon DHPG treatment in WT, but not in KO mice. However, it is quite difficult to see the band at 95 kDa for streptavidin blot (de novo-synthesised PSD-95). In addition, bigger

and smaller bands than 95 kDa were also observed. What are those bands? The blotting needs to be replaced for better presentation.

Following the Reviewer's comment we have replaced the image in Figure 4e with a different one, which is shown as part of the *revised Figure 5* (5d). The other bands (those which don't overlap with signal from the PSD95 antibody) likely correspond to PSD95-associated proteins, as the IP was performed in non-reducing conditions.

(4) Figure5c: A significant decrease in NR2A, but not GluR1 or mGluR5, was detected in the contralateral cortex of WT mice. Given that the spine density change was obvious in the contralateral cortex of WT mice (Fig 5b), NR1A specific decrease seem to be odd. This point should be carefully discussed.

Interestingly, previous studies found similar changes in spine density and NMDARs following whisker deprivation, as indicated in the discussion (refs. 40-42). Furthermore, NR2A expression in the cortex is strongly regulated by synaptic activity and sensory experience (Hoffmann et al., 2000; Philpot et al., 2001; Yoshii et al., 2003; Jaffer et al., 2012) further supporting our findings. We have added these references (refs. 71-74) and comments to the discussion (pg 15).

(5) Figure6a-b: NORT is often used to detect the recognition memory as well as novelty seeking behaviors. In this regard, the longer latency to first contact with the novel object compared to the familiar one is not necessarily indicative of the inability to discriminate between the new and familiar, but it can reflect the decrease in the novelty-seeking traits. Thus, the statement (line 307) that "indicating the inability of BC1 KO mice to discriminate between the two objects" is overstatement. If the authors would like to claim the recognition memory deficit, it should be demonstrated that novelty-seeking trait is intact in KO mice. Related to above, my confusion stems from the sentence in line 316 that "These results support the idea that defects in novel object recognition are not due to deficits in their ability to discriminate novelty, but they are rather specific to the task, possibly involving impaired processing of sensory information". What does this sentence mean? The manuscript needs to be revised in a logical manner. We completely agree with the Reviewer that the latency data does not necessarily indicate ability or inability to discriminate between novel and familiar object, and hence we have replaced latency with data showing the time spent with each object, which are shown now in *new Suppl. Fig. 5*. We have also changed the statement in line 307 to tone down our conclusions. We believe that our data support the notion that novelty-seeking trait is intact in KO mice, as they show preference for novel object in visual NORT as well as for stranger 2 compared to stranger 1 in the social novelty phase of the three-chambered test. We have now modified the sentence in line 316 to better explain our point (pg 11).

(6) Line 337: the authors stated that "our results indicate that BC1 KO mice have deficits in object discrimination, altered social behavior and engage more frequently in self grooming, behavioral phenotypes commonly observed in animal models of ASD, including FXS". Typical ASD model mice also exhibit the altered phenotype of social interaction test (Fig 6d), Marbles test (Fig 6H), and nest-building behaviors (Fig i, j). Since the authors' data are important findings, these must be discussed carefully: how

are BC KO mice different from other ASD model mice, especially FMR KO and CYFIP1 KO mice.

We thank the Reviewer for this important suggestion. We have therefore created a comparative Suppl. Table 1 that includes main findings on Fmr1 KO and Cyfip1+/- mouse models and briefly discussed it in the revised manuscript.

Minor points

(1) The information about the antibodies is insufficient. Please provide enough information so that readers can identify the antibodies that were used.

We are sorry for this brevity. Catalogue number for each antibody has been added in the Method section.

(2) Please provide much detailed information of spine head size analysis.

More information has been included in the Method section.

(3) In some experiments, it is unclear how many mice/neurons/spines were analyzed in each experiment; N values are not sufficiently given. Can the authors please provide these information in the figure legends (or in the new supplemental figure) along with their statistic values?

We apologize for the missing this information. N values and statistics have been added to the figure legends.

Reviewer #3 (Remarks to the Author):

In this manuscript, the authors investigated roles of the brain cytoplasmic (BC1) RNA on cortical functions. Using BC1 knock out mice, they showed the overall increase of spine density, increased spine size, and larger excitatory postsynaptic currents. They also reported altered structural plasticity of the barrel cortex and impaired behaviors of BC1 knock out mice. BC1 has been known to function with the Fragile X Mental Retardation Protein, FMRP, and suggested to be related to the Autistic Spectrum Disorders (ASD). **Thus, the characterization of BC1 in the cortical function is a crucial issue in the field.** Although this manuscript investigated this important issue, the current manuscript is still immature and requires further experiments and discussion. We thank the Reviewer for the comment. We have now added additional data and revised the text accordingly. We hope he/she will be pleased with the revision.

Major

1) The authors showed that while the cortical spine density is increased in the BC1 knock out mice, the frequency of sEPSC is unchanged. The authors should explain this issue.

We thank the Reviewer for highlighting this important point. The discrepancy may be attributed to a different fraction of silent/active synapses between the two genotypes; BC1 KO mice may have more spines with silent or no synapses but similar number of active synapses (and hence similar sEPSC frequency) than WT mice. Interestingly, Harlow et al (2011, Neuron 65(3): 385–398) observed an increased proportion of silent synapses in the barrel cortex of the developing Fmr1-KO mouse. Also, in a model of schizophrenia in which expression of NR2B over NR2A was higher than usual (like in BC1 KO mice), silent synapses persist throughout adulthood (Greenhill & Juczewski et al., 2015, Science 349(6246):424-427).

To verify this intriguing hypothesis (speculative at this stage), a thorough characterization (both at the electrophysiological and biochemical level) would be required which in our opinion is beyond the scope of the present study.

2) The circuit and information flow in the barrel cortex has been well characterized in the previous reports. The authors should examine evoked mEPSC to further elucidate the cortical alteration of BC1 knock out mice.

Following Reviewer's suggestion, we have performed a set of experiments to measure mEPSC in layer 2/3 pyramidal neurons from the barrel cortex. Results are shown below as Figure for the reviewer only. We have not included the figure in the manuscript because this experiment was performed with a limited number of animals (due to logistical problems as a result of our lab relocation to Lausanne).

Quantification of miniature EPSC (mEPSC) amplitude and frequency (n = 20 and 18 neurons from WT and KO mice, respectively). Outliers (open square and closed circle) were disregarded for statistical analysis. No statistical differences between genotypes were observed.

3) In the Figure 3c, the authors showed no change in the frequency of sEPSC of mutant barrel cortex. However, *in vivo* recording exhibited increased spontaneous spikes. The authors should explain this issue.

This is also a good point brought up by the reviewer. The *in vivo* spontaneous spiking we recorded from layer 2/3 pyramidal neurons reflects a combination of excitatory inputs from both cortical and thalamic neurons. In the slice preparation, the active source of sensory excitatory drive from the thalamus is missing, which may account for the observed differences. While we did not measure thalamic activity *in vitro* or *in vivo*, we did observe comparable increases in spine density in BC1 KO neurons from layers 2/3 and 4 suggesting enhanced excitability throughout the thalamo-cortical input pathway. Alternatively, “the increased sEPSC amplitude could result in more cells reaching spiking threshold, thereby leading to enhanced cortical activity” as we stated in the manuscript. In any event, higher frequency of sEPSC does not necessarily translate directly into higher rate of action potentials because it depends on their timing. Thus, synchronization of oscillations or up/down states may be also crucial factors influencing the activity of principal neurons. In this regard, Zhong et al. 2009 (ref. 4) showed several changes in oscillatory properties in BC1 KO mice, which could account for the differences in spontaneous activity *in vivo*.

4) Using *in vivo* recording, the authors should further investigate the functional whisker-barrel map (Stern et al. 2001, Jitsuki et al. 2011). This is important for relating synaptic function, circuit, and behaviors.

We thank the referee for the suggestion. It is indeed an interesting experiment that should be further explored in future studies. However, we believe that it is beyond the scope of this manuscript.

5) In the Figure 4, the authors showed the alteration of NR2B, GluR1, GluR2, and mGluR5 in the mutant barrel cortex. On the other hand, only NR2A is changed in the whisker deprivation condition in the mutant barrel cortex compared to wild type.

As discussed above (Reviewer 2), some studies found similar changes in spine density and NMDARs following whisker deprivation in WT animals, as indicated in the discussion (refs. 40-42). NR2A expression in the cortex is strongly regulated by synaptic activity and sensory experience (Hoffmann et al., 2000; Philpot et al., 2001; Yoshii et al., 2003; Jaffer et al., 2012) further supporting our findings. We have added these references (refs. 71-74) and comments to the discussion.

6) What is a mechanisms underlying the regulation of the expression of NR2A with BC1?

Our data from Fig. 4 and S3 indicate that BC1 RNA does not directly regulate NR2A levels (total or postsynaptic). In our revised manuscript, we further discuss potential underlying mechanisms for whisker deprivation-mediated down-regulation of NR2A expression (i.e. activity- and experience-dependent regulation of NR2A expression, proteasome degradation, etc), and how high PSD95 levels could counteract this effect in BC1 KO mice. In order to confirm this hypothesis, we have performed experiments to determine whether protein degradation at synapses (ubiquitination of proteins in synaptoneurosomes) is affected in BC1 KO mice. Results from this experiment are shown in *new Figure 5c*, and indicate that overall protein degradation is similar between genotypes. Thus, the unbalanced protein synthesis/turnover in BC1 KO mice might explain why NR2A did not change in response to sensory deprivation in BC1 KO mice.

In addition, we have also performed experiments to address how BC1 regulates other GluRs (NR2B, GluR1, mGluR5) by treating WT and BC1 KO synaptoneurosomes with protein synthesis inhibitors. Results from these experiments are shown in *new Fig. 4*, and indicate that BC1 regulates mGluR5 and possibly also NR2B through direct, local translational control. In contrast, BC1 does not seem to regulate GluR1 levels but its localization, possibly as consequence of the increased PSD95 synthesis. Furthermore, we have performed sequence complementarity analysis between BC1 RNA and the mRNA of these GluRs, which confirmed the biochemical data. These bioinformatics analysis are shown in *new Suppl. Fig. 4*.

7) They showed the abnormality of object recognition and social recognition. Since these behaviors could also be regulated by other brain regions than the barrel cortex, the authors should thoroughly test other learning tasks.

A previous study characterized learning and memory in BC1 KO mice (ref. 6). In particular, Lewejohann et al. (2004) used Barnes maze and Morris water maze (memory for a single location), multiple T-maze and complex alley maze (route learning), and radial maze (working memory). In addition to these laboratory tasks, exploratory behavior and spatial memory were assessed under semi-naturalistic conditions in a large outdoor pen. Overall, they did not found any impairment in spatial memory in BC1-deficient mice. Here we have performed the texture novel object recognition test, a memory task specifically dependent on sensory information from the whiskers (Wu et al., 2013, Behav. Brain Res.). Results from this experiment are shown in *new Figure 7* and *revised Suppl Figure 5*.

Minor

1) In the Figures 4a,b, they used actin for the normalization. Why did they use vinculin for the normalization in Figure 5?

There is not a special reason for this, besides that the experiments and western blots were performed by different experimenters over a period of 2-3 years. Both proteins are routinely used as protein loading control in our lab and in other labs as well. Importantly, the levels of these proteins are not different between WT and BC1 KO mice.

2) There is a TypeO at line 255.

Edited

REVIEWERS' COMMENTS:

Reviewer #1 (Remarks to the Author):

In this resubmission of a revised manuscript, the authors have been responsive to previous critiques. Concerns about interpretation of data and genetic background of animals have been adequately addressed. I have no further major concerns. One minor issue: the materials and methods section for SUnSET should be updated for normalization of puromycin immunodetection to Ponceau red staining.

Reviewer #2 (Remarks to the Author):

In this resubmission by Dr. Bagni and colleagues, the authors make significant improvements to the original work and make valiant efforts to address all the comments of this reviewer. I think the manuscript is suitable for Nature Communications.

Reviewer #3 (Remarks to the Author):

I think the reviewer addressed issues raised by the reviewer and the manuscript is not suitable for publication.

Briz et al.

REVIEWERS' COMMENTS:

Reviewer #1 (Remarks to the Author):

In this resubmission of a revised manuscript, the authors have been responsive to previous critiques. Concerns about interpretation of data and genetic background of animals have been adequately addressed. I have no further major concerns. One minor issue: the materials and methods section for SUnSET should be updated for normalization of puromycin immunodetection to Ponceau red staining.

We have added this information (pg 25 of the revised manuscript).

Reviewer #2 (Remarks to the Author):

In this resubmission by Dr. Bagni and colleagues, the authors make significant improvements to the original work and make valiant efforts to address all the comments of this reviewer. I think the manuscript is suitable for Nature Communications.

Reviewer #3 (Remarks to the Author):

I think the reviewer addressed issues raised by the reviewer and the manuscript is now suitable for publication.

I hope he/she meant "now suitable for publication"!